# Pexophagy suppresses ROS-induced damage in leaf cells under high-intensity light

Kazusato Oikawa [1,16], Shino Goto-Yamada[2], Yasuko Hayashi[3], Daisuke Takahashi [4,17], Yoshitaka Kimori[5,6,18], Michitaro Shibata[7], Kohki Yoshimoto[8], Atsushi Takemiya [9], Maki Kondo[1], Kazumi Hikino[10], Akira Kato[3], Keisuke Shimoda[3], Haruko Ueda[11], Matsuo Uemura [4,12], Keiji Numata[13,16], Yoshinori Ohsumi [14], Ikuko Hara-Nishimura [11], Shoji Mano[10,15], Kenji Yamada [2] ✉ & Mikio Nishimura[1,11] ✉

Although light is essential for photosynthesis, it has the potential to elevate intracellular levels of reactive oxygen species (ROS). Since high ROS levels are cytotoxic, plants must alleviate such damage. However, the cellular mechanism underlying ROS-induced leaf damage alleviation in peroxisomes was not fully explored. Here, we show that autophagy plays a pivotal role in the selective removal of ROS-generating peroxisomes, which protects plants from oxidative damage during photosynthesis. We present evidence that autophagy-deficient mutants show light intensity-dependent leaf damage and excess aggregation of ROS-accumulating peroxisomes. The peroxisome aggregates are specifically engulfed by pre-autophagosomal structures and vacuolar membranes in both leaf cells and isolated vacuoles, but they are not degraded in mutants. ATG18a-GFP and GFP-2×FYVE, which bind to phosphatidylinositol 3-phosphate, preferentially target the peroxisomal membranes and pre-autophagosomal structures near peroxisomes in ROS-accumulating cells under high-intensity light. Our findings provide deeper insights into the plant stress response caused by light irradiation.

Photosynthesis in plants converts light energy to chemical energy and is accompanied by photorespiration, which involves peroxisomes, mitochondria, and chloroplasts[1]. Photorespiration is essential for plant survival under high-intensity light and prevents photoinhibition, which damages photosynthetic machinery owing to excess reactive oxygen species (ROS) accumulation[2,3]. Thus, it is essential to understand how excess ROS are degenerated to protect plants from oxidative damage during photosynthesis under excess light.

Plants have diverse mechanisms to prevent high ROS accumulation under various conditions[4–10], and the relationship between ROS and autophagy has been reported in the previous studies[11–14]. ROS accumulation in peroxisomes inhibits catalase (CAT) activity that detoxifies hydrogen peroxide, leading to the oxidation of peroxisomes[5,9,10]. We have previously shown that oxidatively damaged peroxisomes are accumulated in autophagy-deficient mutants[11,14].

A set of *autophagy* (*ATG*) genes has been discovered to be involved in the specific degradation of peroxisomes, namely pexophagy in yeasts and animals[15–19]. ATG proteins initiate autophagy by forming pre-autophagosomal structures (PAS) on vacuolar membranes containing phosphatidylinositol 3-phosphate (PtdIns3P) adjacent to degraded peroxisomes in yeast[20–22]. Subsequently, a membrane structure called the phagophore extends from the PAS to cover peroxisomes by incorporating phosphatidylethanolamine (PE)-conjugated ATG8 (ATG8-PE) and then fuses with vacuoles for degradation[18,19,22]. Most ATG proteins are highly conserved in yeasts, animals, and plants[15–19]. However, it is unclear whether pexophagy in plants is the same as that in yeasts and animals because homologues of

key factors for pexophagy in yeasts, namely PpAtg30 and ScAtg36, are absent in plants. Moreover, no direct evidence exists for the selective degradation of peroxisomes by pexophagosomes in plant cells[23].

Here, we investigate the cell-structural mechanism underlying the autophagy-dependent degradation of ROS-accumulated peroxisomes to determine the pexophagosome formation in Arabidopsis leaves. Furthermore, we examined the impact of pexophagy deficiency on ROS accumulation-induced leaf damage caused, which is accompanied by the accumulation of damaged catalases in peroxisomes. Our finding indicates a massive contribution of pexophagy in protecting plants from excess light-induced oxidative damage during photosynthesis.

## Results

### A difference in the pattern of peroxisome aggregation in leaf mesophyll cells between *atg2* and *atg7* mutants

We previously isolated *peup1/atg2* (*atg2(p1)*), *peup2/atg18a* (*atg18a(p2)*), and *peup4/atg7* (*atg7(p4)*) mutants defective in pexophagy from ethylmethane sulfonate (EMS)-mutagenised GFP-PTS1 (wild type) lines[11]. Since the genes responsible for *peups* were *ATGs*, we obtained T-DNA insertion mutants of *atg2*, *atg5*, *atg7*, *atg18a*, and *atg9*[11,14]. The *atg* mutants other than *atg9*, in which peroxisomes were visualised by introducing GFP-PTS1, showed peroxisome aggregation in leaf mesophyll cells. However, the peroxisome-aggregation patterns were different between the *atg* mutants at 100 μmol m$^{-2}$ s$^{-1}$ (Fig. 1a–d, Supplementary Fig. 1, and Supplementary Movies 1–3). The number of peroxisomes and peroxisome aggregates in *atg7* cell was higher than that in *atg2* and the other *atg* mutants cell (Fig. 1b, Supplementary Fig. 1), while the size of peroxisome aggregates (Fig. 1c) was lower in *atg7* than that in *atg2*. The frequency of the cell containing the peroxisome aggregates is similar between *atg2* and *atg7* (Fig.1d).

Each allele of the *atg* mutants revealed the same results (Supplementary Fig. 1).

Unlike non-selective, starvation-induced autophagy that recycles nutrients such as carbon and nitrogen[15–17], ROS-induced autophagy in mammals selectively degrades damaged organelles[12,24]. To further obtain insight into the mechanism underlying ROS-promoted pexophagy in plants exposed to light, we first assessed plant growth of *atg2(p1)* and *atg7(p4)* mutants under different light intensities (50, 100, and 200 μmol m$^{-2}$ s$^{-1}$). The *atg2(p1)* and *atg7(p4)* leaves were more damaged than the wild-type leaves (Supplementary Fig. 2a). With increases in light intensity, the mutants showed reduced chlorophyll (Supplementary Fig. 2b) and photosynthetic efficiency (Supplementary Fig. 2c). Damage to *atg7(p4)* leaves was more extensive than that to *atg2(p1)* leaves under light; therefore, we mainly focused on *atg7(p4)*. ROS accumulation was higher in the damaged *atg7(p4)* leaves under 100 and 200 μmol m$^{-2}$ s$^{-1}$ light, as revealed by nitroblue tetrazolium (NTB) staining (Supplementary Fig. 2d, e). Electron microscopy analyses of *atg7(p4)* revealed abnormal high-density regions in the peroxisomes (dark-grey regions; Supplementary Fig. 3a, b). These dark-grey regions contained large amounts of catalase (Supplementary Fig. 3c–f) similar to those in *atg2(p1)* and *atg5* mutants[11,14]. Immunoblotting confirmed the high catalase accumulation in *atg7(p4)* under 200 μmol m$^{-2}$ s$^{-1}$ light (Supplementary Fig. 4); thus, catalase might be non-functional also in *atg7(p4)* leaf cells. In the leaves of the *atg7(p4)* mutant, tubular structures from peroxisomes, namely peroxules, were formed, suggesting that leaves accumulated a high level of ROS[25] (Supplementary Fig. 5a, b and Supplementary Movie 4). Interestingly, we observed that chloroplasts in the *atg7(p4)* mutant ingested some peroxisomes (Supplementary Fig. 5c, d), which was similar to chloroplast behaviour in ROS-accumulating cells[26]. Detailed analysis with an electron microscope revealed that some peroxisomes were ingested by curved chloroplast membranes (Supplementary Fig. 5e). These results suggest that the *atg7(p4)* mutant accumulates high ROS and damaged catalase levels in peroxisomes, resulting in plant growth inhibition.

### ATG18a preferentially targets leaf peroxisomes in light-adapted cells

ATG18a plays a role in autophagosome formation and the degradation of oxidised proteins in *Arabidopsis*[27], indicating that ATG18a targets damaged organelles. To examine ATG18a localisation during pexophagy in plants, we assessed the intracellular distribution of ATG18a-GFP in wild type as well as *atg2(p1)* and *atg7(p4)* mutants in which peroxisomes were visualised with red fluorescence protein-fused peroxisomal targeting signal 1 (RFP-PTS1; Fig. 2a). The ATG18a-GFP-containing structures localised to peroxisomes, although they are rarely observed in wild-type plants (Fig. 2a–d). We discovered that numerous cells accumulated ATG18a-GFP structures on peroxisomes in the *atg2(p1)* as well as *atg7(p4)* mutants (Fig. 2b) and that 30–40% of the total peroxisomes, especially aggregated peroxisomes, had the ATG18a-GFP structures (Fig. 2c). Furthermore, approximately 80% of the ATG18a-GFP structures localised to peroxisomes in wild-type and mutant cells (Fig. 2d), suggesting that ATG18a preferentially targets the peroxisome.

We have previously shown that ATG8 accumulates near the peroxisome aggregates in *atg2(p1)*[11,14]. To examine whether ATG18 and ATG8 target the same peroxisome aggregate in *atg2(p1)*, we transiently expressed CFP-ATG8e in *atg2(p1)* and *atg7(p4)* expressing ATG18a-GFP. The result showed that CFP-ATG8e and ATG18a-GFP are colocalised to the same peroxisome aggregate (Supplementary Figs. 6, 7), revealing that ATG18a recognises oxidized peroxisomes to be degraded.

Immunoblotting showed ATG18a-GFP and catalase in the insoluble fraction of *atg2(p1)* and *atg7(p4)* (Supplementary Fig. 8). Most of the ATG18a-GFP appeared as dot structures, while a few were cup or ring structures in *atg2(p1)* and *atg7(p4)* mutants at 100 μmol m$^{-2}$ s$^{-1}$ light intensity (Fig. 2a, e–g and Supplementary Fig. 9a, c and Supplementary Table 1). We tracked single peroxisomes by time-lapse imaging, and then the average image of RFP-PTS1 and ATG18a-GFP (Fig. 2h) was generated using a morphological image processing tool[28]. The image revealed a ring structure of ATG18a-GFP surrounding the peroxisome in *atg7(p4)*. Time-lapse imaging also showed that ATG18a-GFP gradually surrounded peroxisomes in wild type and *atg7(p4)*, but not in *atg2(p1)* (Supplementary Fig. 10 and Supplementary Movies 5–7). Furthermore, *atg2(p1)* had fewer ring structures compared to *atg7(p4)* (Fig. 2f). In the *atg2(p1)* and *atg7(p4)* mutant, more than half of ATG18a-GFP fluorescence was recovered within 60 s after photobleaching, indicating that ATG18a-GFP rapidly accumulates at the peroxisome aggregates (Supplementary Figs. 11, 12 and Supplementary Movies 8, 9).

To examine whether ATG18a interacts with other proteins, we conducted an immunoprecipitation of ATG18a-GFP followed by protein mass spectrometry. The result shows that various proteins of chloroplasts, peroxisomes, and mitochondria were co-immunoprecipitated with ATG18a-GFP in *atg2(p1)* (Supplementary Fig. 13 and Supplementary Table 2). Peroxisome proteins such as catalases (CAT1, CAT2, and CAT3), heat shock protein 70s (HSP70s), and RuBisCO-related proteins were abundantly present. We obtained the number of proteins localised to each organelle from two databases (PPDB, http://ppdb.tc.cornell.edu/dbsearch/subproteome.aspx; and SUBA4, http://suba.live) and calculated the recovery rate. The peroxisome proteins were more efficiently recovered compared to the mitochondria and chloroplast proteins (Supplementary Fig. 13b); thus, suggesting that numerous ATG18a proteins directly or indirectly bind to peroxisomes or peroxisomal proteins.

### PtdIns3P accumulates on leaf peroxisomes in light-adapted cells

Arabidopsis ATG18a has a PtdIns3P-binding motif similar to yeast ATG18 and ATG21[20,29,30]. To validate whether peroxisomes marked by ATG18a-GFP exhibit the PtdIns3P motif, we examined the intracellular distribution of GFP-2×FYVE, a reliable PtdIns3P-binding marker[31,32], in

 

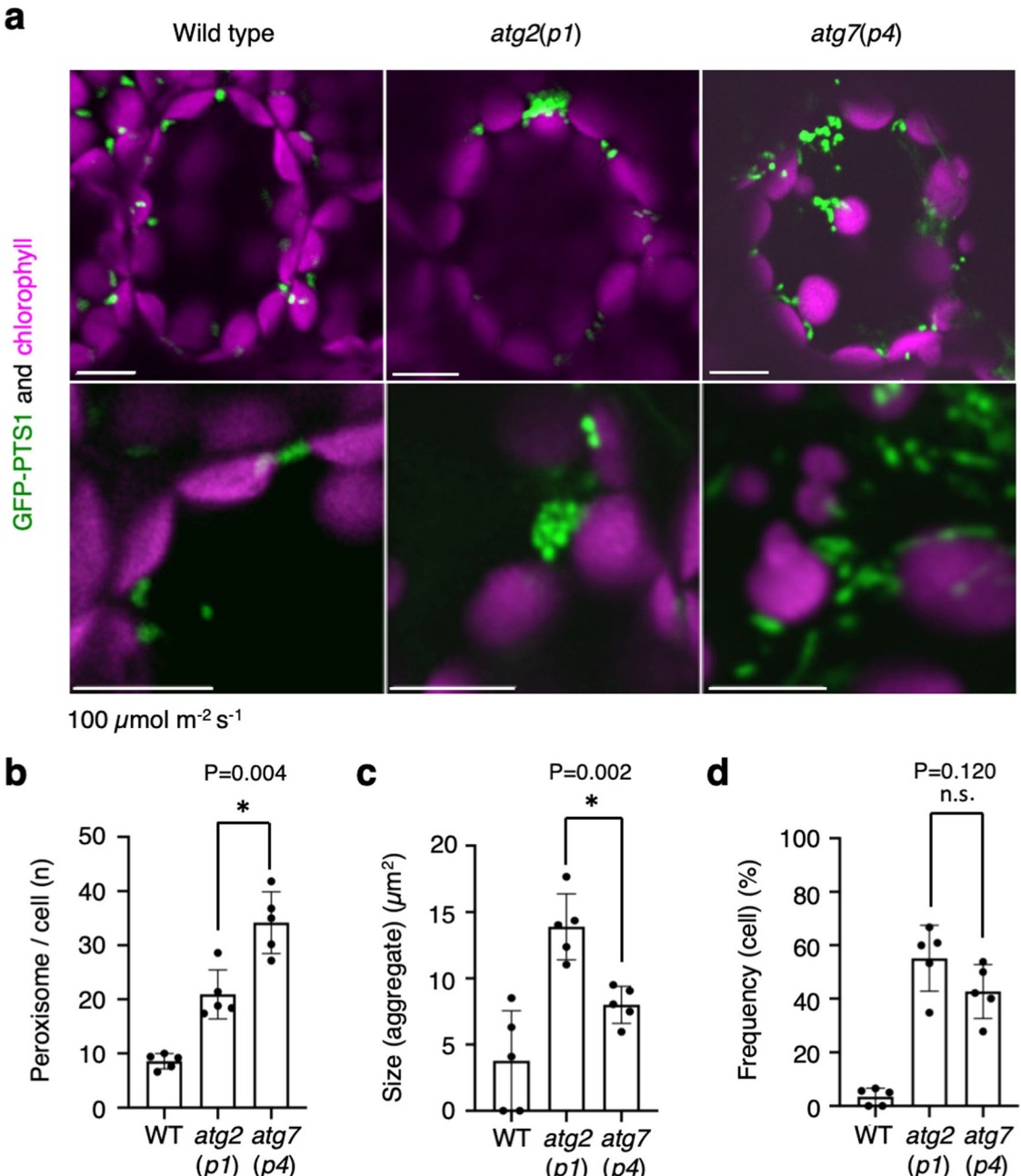

**Fig. 1 | Difference in phenotypes of peroxisome aggregates between *atg2(p1)* and *atg7(p4)*. a** Peroxisomes (GFP-PTS1, green) and chloroplasts (auto-fluorescence, magenta) in leaf mesophyll cells of wild type (WT), *atg2(p1)*, and *atg7(p4)*. The bottom images are enlarged images of peroxisome aggregates. Images are obtained from the surface to middle depth region of 3-week-old plants cultured on an agar plate containing ½ MS with 1% sucrose under normal-intensity (100 µmol m$^{-2}$ s$^{-1}$) white-light conditions. Scale bars, 10 µm. **b** Peroxisome number per cell. **c** The average size of peroxisome aggregates. **d** Frequency of cells containing the peroxisome aggregates. **a–d** Biologically independent leaf cells of WT ($n$ =127), *atg2(p1)* ($n$ =101), and *atg7(p4)* ($n$ =94) are tested. The graphs show a summary of at least five independent experiments. The error bars indicate mean ± standard deviation. Asterisks indicate significant differences between *atg2* and *atg7* (*$P$ < 0.01) (**b**, **c**) and n.s. indicates not significant differences ($P$ > 0.05) (**d**) in the two-sided Student's *t*-test.

*atg2(p1)* and *atg7(p4)* mutants. We discovered that GFP-2×FYVE showed similar trends in the fluorescence pattern (Fig. 3a, e–g and Supplementary Fig. 14 and Supplementary Movies 10, 11) and frequency of peroxisome targeting (Fig. 3b, c) as ATG18a-GFP (Fig. 2a–c, e–g and Supplementary Fig. 10 and Supplementary Movies 5, 7). These findings indicate that PtdIns3P accumulates on the membrane of peroxisomes and suggest that the ATG18a-GFP recognises PtdIns3P on the membrane during its accumulation in peroxisomes. We further conducted a lipid-binding test for ATG18a-GFP protein in transgenic plants. We confirmed the ability of ATG18a-GFP protein to bind to PtdIns3P (Supplementary Fig. 15a). We examined the localisation of GFP-2×FYVE and ATG18a-GFP in detail with immunoelectron micro-scopic analysis using anti-GFP antibodies (Supplementary Figs. 15b, 16) and observed that GFP-2×FYVE and ATG18a-GFP were localised both on the peroxisome periphery and adjacent to peroxisomes where the phagophore is expected to reside. These localisations were similar to that of ATG8 analysed using an anti-ATG8 antibody (Supplementary Figs. 17). Detailed analysis using electron microscopy revealed ER- and autophagosome-like structures adjacent to the high-density area in *atg2(p1)* mutant peroxisomes (Supplementary Fig. 15b, c).

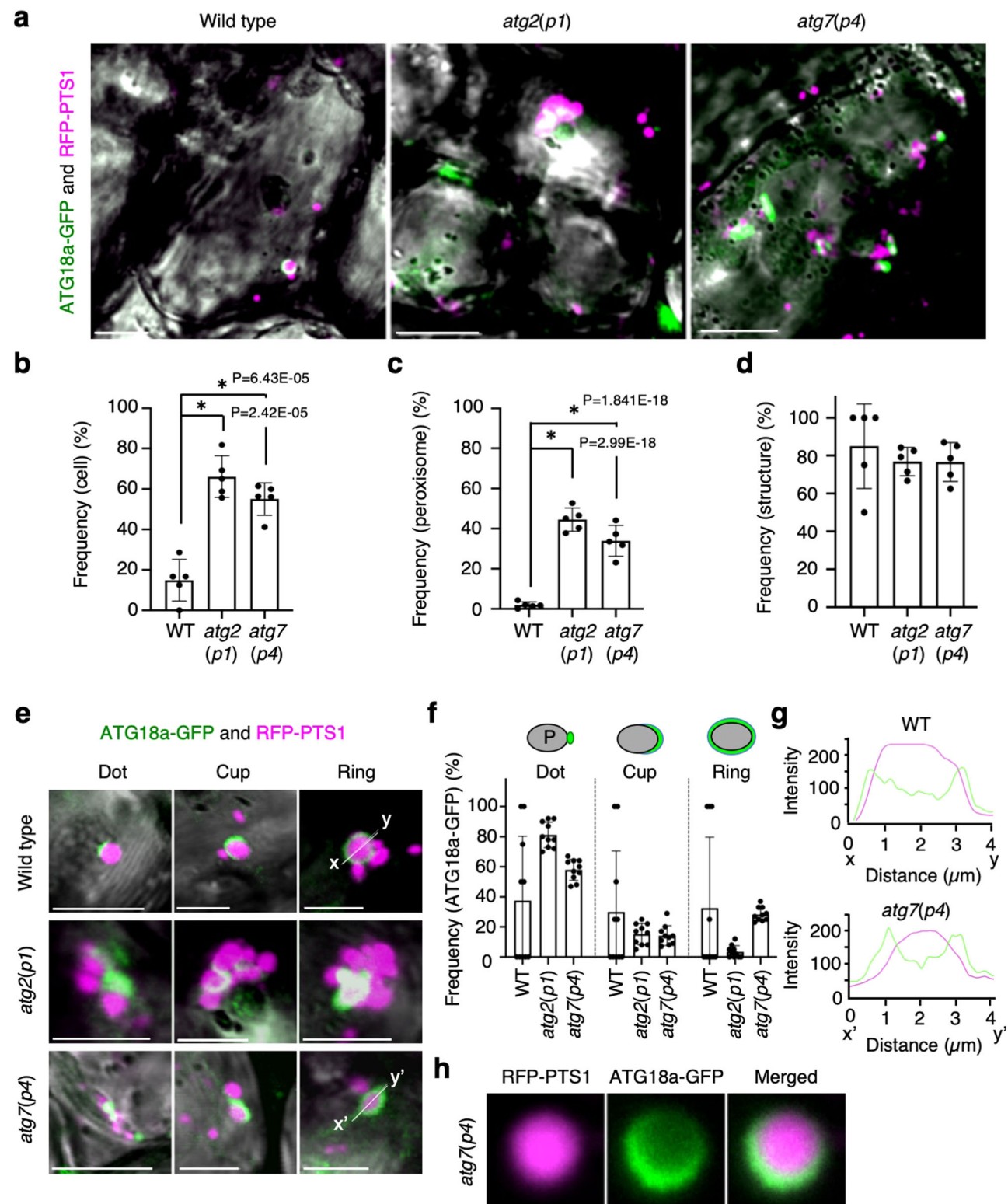

The ring structure was also noted in GFP-2×FYVE (Fig. 3e–g and Supplementary Fig. 9b, d), although to a lesser extent than that observed in ATG18a-GFP (Figs. 2f, 3f and Supplementary Fig. 9a, c and Supplementary Table 1), suggesting that GFP-2×FYVE mainly functions at the initial step of ring formation in the pexophagy process. Approximately half of the dot structures of GFP-2×FYVE did not target peroxisomes (Fig. 3d and Supplementary Table 3), indicating that there are other accumulations of PtdIns3P in the cell. GFP-2×FYVE was also localised to peroxisome aggregates in the *atg18a(p2)* mutant

(Supplementary Figs. 18a, b), similar to the *atg2(p1)* and *atg7(p4)* mutants; thus, suggesting that PtdIns3P accumulation on peroxisomes precedes the action of ATG18a, ATG2, and ATG7 during pexophagy. This was supported by the evidence that wortmannin[16,18,21,32], a phosphoinositide 3-kinase inhibitor, disturbed the subcellular localisation of ATG18a-GFP and GFP-2xFYVE on peroxisome aggregates in *atg2(p1)*, *atg18a(p2)*, and *atg7(p4)* (Supplementary Fig. 19).

The autophagic process associated with the engulfing of an object to be degraded in the cytosol with the isolation membrane (formed by

**Fig. 2 | ATG18a-GFP targets peroxisomes. a** Confocal microscope images of peroxisomes (RFP-PTS1, magenta) and ATG18a-GFP (green) in leaf mesophyll cells of wild-type (WT), *atg2(p1)*, and *atg7(p4)* cells. The fluorescent images were merged with bright-field images. Images were obtained from the surface to middle depth region of 3-week-old plants cultured on an agar plate containing ½ MS with 1% sucrose under normal-intensity (100 μmol m⁻² s⁻¹) white-light conditions. Scale bars, 10 μm. **b** Ratio of the cells containing peroxisomes bound by ATG18a-GFP to total cells. More than 100 cells were tested. **c** Ratio of peroxisomes bound by ATG18a-GFP to total peroxisomes. **d** Ratio of ATG18a-GFP structures bound to peroxisomes to total ATG18a-GFP structures. More than 250 peroxisomes were tested in (**c**, **d**). The error bars indicate mean ± standard deviation (five biological

replicates), and asterisks indicate significant differences between WT and *atg2(p1)* or *atg7(p4)* (*$P < 0.01$, two-sided Student's *t*-test) (**b–d**). **e** Types of ATG18a-GFP localisation on peroxisomes. The structures are categorised into three types: dot, cup, and ring. Scale bars, 2 μm. **f** Frequency of the types of ATG18a-GFP localisation in *atg2(p1)* and *atg7(p4)*. The error bars indicate mean ± standard deviation ($n = 10$ biologically independent replicates). **g** Plot profiles of RFP (magenta) and GFP (green) fluorescence on lines (x−y and x′−y′) in (**e**). **h** An averaged image of fluorescence during time-lapse imaging of peroxisome (RFP-PTS1, magenta) surrounded by ATG18a-GFP (green) for 300 s. Scale bars, 1 μm. The representative images in **a**, **e**, **h** show a summary of at least five independent experiments.

assemblies of ATG8 and other ATG factors) and its subsequent transportation to the vacuole is technically referred to as macroautophagy[18]. Our findings suggest that ATG18a recognises PtdIns3P on the membrane of damaged peroxisomes or PAS associated with damaged peroxisomes and facilitates the segregation and degradation of peroxisome via macroautophagy, i.e., macropexophagy, in plants.

## High-intensity light causes leaf damage and high levels of peroxisomal aggregation in autophagy-defective mutants

Next, we investigated the effect of high-intensity light (1000 μmol m⁻² s⁻¹) damage on leaves of *atg7(p4)* mutants (Fig. 4) and T-DNA insertion mutants for *atg2*, *atg5*, *atg7*, and *atg9* (Supplementary Fig. 20a). Leaf damage and chlorophyll degradation were observed in *atg2*, *atg5* and especially *atg7* (Fig. 4a and Supplementary Fig. 20). Remarkably, the large aggregates of peroxisomes were induced in leaf mesophyll cells of *atg2, atg5,* and *atg7*, mostly at the cell bottom (Fig. 4b and Supplementary Fig. 21a). The frequency and size of peroxisome aggregates in *atg2, atg5,* and *atg7* under high-intensity light condition was two to three times greater than that under normal light conditions (100 μmol m⁻² s⁻¹; Fig. 4c, d and Supplementary Figs. 21b, 22a,b). When subjected to high-intensity light, the accumulation of insoluble catalase was higher in the *atg2, atg5,* and *atg7* than that in the wild type (Fig. 4e–h and Supplementary Figs. 21c, 22c–e); thus, suggesting that the accumulation of inactive catalases leads to peroxisome aggregates in the mutants. We overexpressed GFP-CAT2 or RFP-CAT2 to recover catalase activity and discovered that CAT2 fusion overexpression suppressed the increase in peroxisome numbers and their aggregation in the *atg2* mutant (Supplementary Fig. 23). To examine vacuolar peroxisome degradation by pexophagy under high-intensity light conditions, we inhibited vacuolar H⁺-ATPase using concanamycin A (ConA) to stop vacuolar hydrolytic activity in wild-type leaf mesophyll cells (Supplementary Fig. 24). The ConA-treated cells increased the accumulation of undegraded-peroxisomes in the vacuole under high-intensity light conditions (Supplementary Fig. 24a–c and Supplementary Movies 12–15). These results indicated that pexophagy was facilitated in high-intensity light.

Furthermore, we observed an increase in the number of mitochondria, which gathered to peroxisome aggregates in *atg7(p4)* under high-intensity light conditions, suggesting that mitophagy was also suppressed (Supplementary Fig. 25a–c). Additionally, mitochondrial proteins serine hydroxymethyltransferase (SHMT) and cytochrome c oxidase 2 (COXII) were slightly increased in *atg7(p4)* under high-intensity light conditions (Supplementary Fig. 25d–i). These results suggest that ATG7 plays multiple roles in the degradation of damaged mitochondria as well as that of peroxisomes in leaves undergoing photosynthesis.

## Vacuolar membranes surround large peroxisome aggregates with ATG18a and PtdIns3P in high-intensity light

To further investigate whether the large peroxisome aggregates that are induced under high-intensity light are degraded by autophagy, we

focused on the subcellular localisation of ATG18a-GFP and GFP-2×FYVE in 1000 μmol m⁻² s⁻¹ light-adapted leaf cells of *atg2(p1)* and *atg7(p4)* mutants (Fig. 5a–d). The results indicate that ATG18a-GFP and GFP-2×FYVE preferentially targeted the large aggregates of peroxisomes in *atg2(p1)* and *atg7(p4)* mutants. Moreover, peroxisome aggregates in *atg7(p4)* mutants were largely enveloped by ATG18a-GFP (Fig. 5a, b and Supplementary Table 4) and the frequency of these peroxisome aggregates was approximately 43% in *atg7(p4)* and 11% in *atg2(p1)* (Fig. 5c and Supplementary Table 4). The size of peroxisome aggregates enveloped by ATG18a-GFP in *atg7(p4)* was ~34 μm², which was eight times greater than that observed in wild type and six times greater than that in *atg2(p1)* (Fig. 5d). In contrast, both the frequency and size of peroxisome aggregates enveloped by GFP-2×FYVE were smaller than those enveloped by ATG18a-GFP in all tested lines (Fig. 5c, d). The analysis of fluorescent intensities in the aggregates confirmed that ATG18a-GFP co-localised with the large aggregates of peroxisomes (Fig. 5e).

We further investigated the relationship between vacuolar membranes and peroxisomes in wild type and *atg7(p4)* using a vacuolar membrane marker, namely Venus-VAM3, in high-intensity light (Fig. 5f, g). We discovered that the vacuolar membrane structures depressing toward the interior, dubbed vacuolar cavities, frequently surrounded peroxisome aggregates in *atg7(p4)* (Fig. 5f, g). The frequency of vacuolar cavities was similar to that of peroxisome aggregates with ATG18a-GFP (Fig. 5c, h and Supplementary Tables 4, 5); thus, suggesting a similar developmental mechanism between the two structures. The cells with vacuolar cavities were three times more abundant in *atg7(p4)* than in wild type (Fig. 5h). The size of vacuolar cavities containing peroxisome aggregate was also larger in *atg7(p4)* than in wild type (Fig. 5i). Compared to wild type, *atg7(p4)* showed a higher frequency of the vacuolar cavities surrounding peroxisomes and peroxisome aggregates under high-intensity light conditions (Supplementary Fig. 26 and Supplementary Movies 16, 17). The frequency of vacuolar cavities surrounding peroxisome aggregates in *atg7(p4)* was approximately 40% (Fig. 5j). These results suggest that vacuolar cavities surrounding peroxisome aggregates are involved in the process of pexophagy following high-intensity light treatment.

To understand the peroxisome aggregate and vacuolar membrane association, we isolated vacuoles from transgenic plants expressing RFP-PTS1 and Venus-VAM3 after exposure to high-intensity light treatment that enhances pexophagy (Fig. 6 and Supplementary Figs. 27–29). RFP fluorescence was observed on the vacuolar surface in the *atg7(p4)*, whereas the fluorescence was observed in the vacuolar lumen of wild-type cells (Fig. 6a–h, Supplementary Figs. 27–29 and Supplementary Movies 18–20). These findings suggest that peroxisome aggregates are attached to the surface of isolated vacuoles in *atg7(p4)* while being further assimilated into vacuoles in the wild type (Fig. 6a–h). The enlarged and time-lapse images revealed that peroxisomes were surrounded by the vacuolar membrane and assimilated into the vacuole in the wild type (Fig. 6e, f, Supplementary Fig. 27a–d, and Supplementary Movies 18–20). We analysed the frequency of RFP fluorescence surrounded with Venus-VAM3 and discovered that ~25% of the peroxisome aggregates were surrounded by the vacuolar

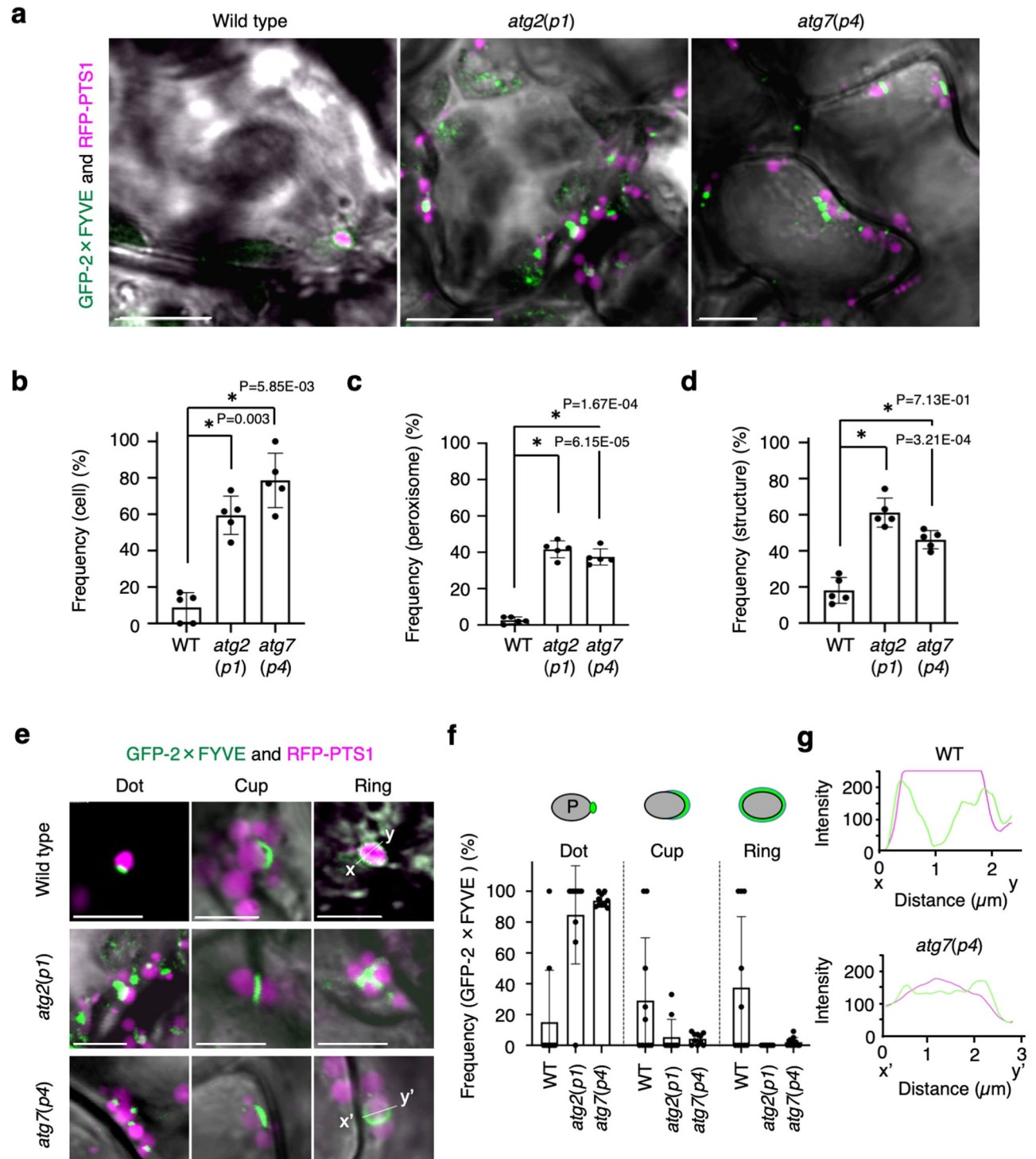

**Fig. 3 | GFP-2×FYVE targets peroxisomes. a** Confocal microscope images of peroxisomes (RFP-PTS1, magenta) and GFP-2×FYVE (green) in WT, *atg2(p1)*, and *atg7(p4)* leaf mesophyll cells. Images were obtained from the surface to middle depth region of 3-week-old plants cultured on an agar plate containing ½ MS with 1% sucrose under normal-intensity (100 µmol m$^{-2}$ s$^{-1}$) white-light conditions. Scale bars, 10 µm. **b** Ratio of the cells containing peroxisomes bound by GFP-2×FYVE to total cells. More than 100 cells were tested. **c** Ratio of the peroxisomes bound by GFP-2×FYVE to total peroxisomes. **d** Ratio of GFP-2×FYVE structures bound to peroxisomes to total GFP-2×FYVE structures. More than 250 peroxisomes were

tested in (**c**, **d**). The error bars indicate mean ± standard deviation (five biological replicates), and asterisks indicate significant differences between WT and *atg2(p1)* or *atg7(p4)* (*P* < 0.01, two-sided Student's *t*-test) (**b**–**d**). **e** Types of GFP-2×FYVE localisation on peroxisomes. The structures are categorised into three types: dot, cup, and ring. Scale bars, 2 µm. **f** Frequency of the types of GFP-2×FYVE localisation in *atg2(p1)* and *atg7(p4)*. The error bars indicate mean ± standard deviation (*n* = 10 biologically independent replicates). **g** Plot profiles of RFP (magenta) and GFP (green) fluorescence from lines (*x*–*y* and *x'*–*y'*) in (**e**). The representative images in **a**, **e** show a summary of at least five independent experiments.

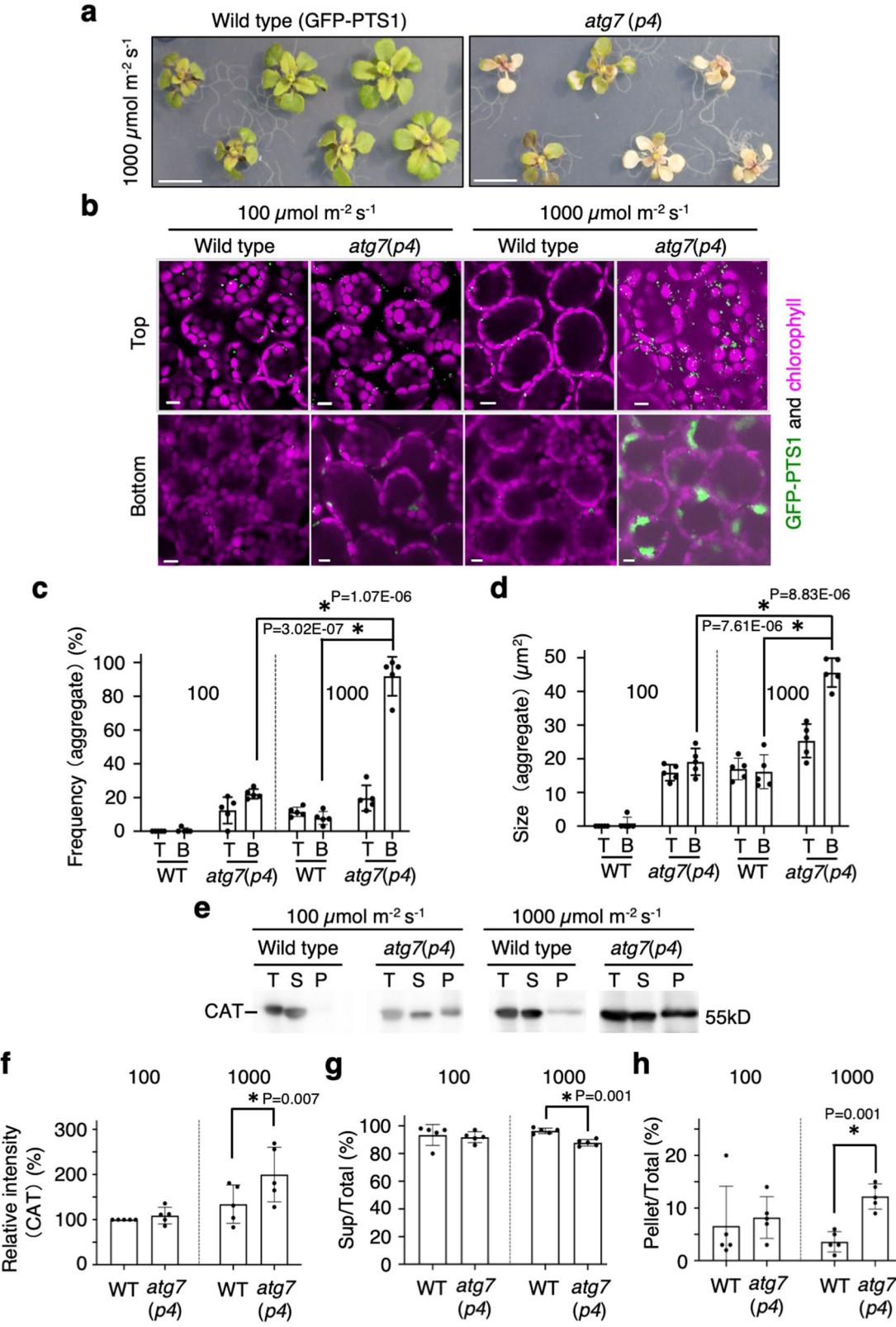

membrane in the vacuoles isolated from the *atg7(p4)* mutant (Fig. 6i). This suggests that although the vacuolar membrane surrounded the peroxisome aggregate, assimilation did not occur in *atg7(p4)*, which is consistent with reduced RFP-fluorescence observed in the vacuolar lumen of the mutant (Supplementary Fig. 27e). Technically, autophagy in which a target for degradation is directly enclosed by the vacuolar membrane and taken up into the vacuole is referred to as

microautophagy. Therefore, peroxisome aggregates seem to be degraded via microautophagy, i.e., micropexophagy, under high-light irradiation conditions. We also examined ATG18a localisation to the isolated vacuoles in plants expressing ATG18a-GFP and RFP-PTS1. We observed co-localisation of ATG18a-GFP and RFP-PTS1 on the vacuoles isolated from *atg7(p4)* (Supplementary Figs. 28, 29 and Supplementary Movies 21, 22), indicating that the peroxisomes on the vacuolar

**Fig. 4 | Formation of large aggregates of peroxisomes in high-intensity light-damaged leaves of _atg7(p4)_ plants. a** Plant phenotype of WT and _atg7(p4)_ under high-intensity light (1000 μmol m$^{-2}$ s$^{-1}$) with LED equipment (blue and red light) for 16 h on agar plates (_n_ = 3 technically independent replicates). Three-week-old plants are cultured on an agar plate containing ½ MS with 1% sucrose under normal-intensity (100 μmol m$^{-2}$ s$^{-1}$) white-light conditions used for the test. Scale bars, 2 cm. **b** Peroxisomes (GFP-PTS1, green) and chloroplasts (autofluorescence, magenta) in WT and _atg7(p4)_ leaf mesophyll cells after adaptation to low (100 μmol m$^{-2}$ s$^{-1}$) and high-intensity light (1000 μmol m$^{-2}$ s$^{-1}$) produced by an LED (blue and red light) for 16 h. Scale bars, 10 μm. **c, d** Frequency of cells containing peroxisome aggregates (**c**) and the size of peroxisome aggregates (**d**) under low and high-intensity light. The region from the top to the middle depth (T) and from the middle to the bottom (B) were observed. More than 165 cells were tested. The error bars indicate mean ± standard deviation (_n_ = 5 biologically independent replicates), and asterisks indicate significant differences between WT and _atg7(p4)_ (*_P_ < 0.01, two-sided Student's _t_-test) in (**c**, **d**). **e** Immunoblot analysis of CAT in total (T), supernatant (S), and pellet (P) fractions of leaf extracts from WT and _atg7(p4)_ plants grown under low and high-intensity light conditions (_n_ = 3 technically independent replicates). **f–h** Relative intensity of anti-catalase antibody signals on immunoblots (**f**), and ratios of them in the supernatant to total extract (**g**) and pellet to total extract (**h**) was calculated using ImageJ. The error bars indicate mean ± standard deviation and asterisks indicate significant differences between WT and _atg7(p4)_ (*_P_ < 0.01, two-sided Student's _t_-test) (_n_ = 5 biologically independent plants) in (**f–h**).

membranes are in the intermediate part of the autophagic pathway. These results suggested that ATG18 may accumulate during the formation of vacuolar membrane cavities surrounding peroxisome aggregates and that ATG7 is involved in the assimilation of these structures into the vacuoles through the microautophagy process under the high-intensity light condition.

### Large peroxisome aggregates promote ROS formation under high-intensity light

We investigated the accumulation of ROS in leaves exposed to high-intensity light (1000 μmol m$^{-2}$ s$^{-1}$). Nitro blue tetrazolium (NBT) staining showed that leaves of both _atg2(p1)_ and _atg7(p4)_ mutants accumulated more ROS compared to that observed in wild-type leaves (Supplementary Fig. 30a, b). Next, we examined the accumulation of ROS in peroxisomes using 27'-dichlorodihydrofluorescein diacetate (H$_2$-DCF-DA)[33,34] (Fig. 7a, b and Supplementary Fig. 30c–f). H$_2$-DCF fluorescence was detected inside mutant peroxisomes with approximately two-fold higher intensity than that associated with wild-type peroxisomes under high-intensity light conditions and elevated compared to those measured under normal-intensity light conditions (Fig. 7b and Supplementary Fig. 30c, d). We discovered that some peroxisomes in wild-type leaf cells and approximately 60% of peroxisome aggregates in the mutant-leaf cells were specifically stained with H$_2$-DCF (Supplementary Fig. 30c, e). The average fluorescence intensity in peroxisomes was two- to three-fold higher than that in chloroplasts, which was especially prominent in the mutants (Supplementary Fig. 30f). Consistently, a small number of chloroplasts were targeted by the GFP-2×FYVE and ATG18a-GFP in the _atg2(p1)_ and _atg7(p4)_ mutants compared to peroxisomes (Supplementary Fig. 31). We concluded that ROS accumulates at high levels in peroxisome aggregates of _atg2(p1)_ and _atg7(p4)_ mutants grown under high-intensity light.

To ascertain the contribution of autophagy in suppressing ROS in leaf cells exposed to high-intensity light, we examined transgenic plants overexpressing ATG18a-GFP. The ATG18a-GFP overexpressing plants showed a reduction in ROS accumulation and an increase in chlorophyll content in the leaf cells (Supplementary Fig. 32); thus, suggesting that the ATG18a overexpression reduces ROS accumulation and chloroplast damage. We further examined the effect of salt stress[35,36] on pexophagy to determine whether the other types of autophagy-inducing stress mimic high-light-induced pexophagy. The salt stress slightly modulated peroxisome aggregation, but to a lesser extent, despite the high accumulation of ROS in wild-type and mutants (Supplementary Figs. 33, 34). The results suggest that salt stress is inefficient compared to high-light stress in pexophagy induction.

In summary, high-intensity light induces leaf damage in _atg2(p1)_ and _atg7(p4)_, and this is accompanied by a remarkable increase in the size and frequency of peroxisome aggregates. The peroxisome aggregates accumulate high levels of ROS with inactive catalase and are recognised by GFP-2×FYVE, CFP-ATG8, and ATG18a-GFP. The overexpression of catalase suppresses the increase in peroxisome aggregate number, while the overexpression of ATG18a-GFP reduces the accumulation of ROS in both the cytosol and peroxisome under high-intensity light conditions. In contrast, salt stress has little effect on pexophagy under normal-intensity light conditions. These results suggest that high-intensity light induces high levels of ROS in peroxisomes due to the impairment of the catalase activity, resulting in the aggregation and subsequent degradation of peroxisomes in the vacuole via macro- and micropexophagy.

## Discussion

We discovered light-induced leaf damage in _atg2(p1)_ and _atg7(p4)_ mutants (Fig. 4 and Supplementary Figs. 2, 20). Increases in light intensity increased leaf damage, suggesting the involvement of photosynthesis. As high-intensity light induces ROS accumulation via photosynthesis, we speculated that light-induced ROS accumulation caused leaf damage in the mutants. In line with our hypothesis, we observed light-dependent ROS accumulation in the leaves of _atg2(p1)_ and _atg7(p4)_ mutants (Fig. 7a, b and Supplementary Fig. 30), indicating that these mutants generate higher levels of ROS compared to wild-type under high-intensity light conditions. These results suggest that high levels of ROS in _atg7(p4)_ induce the formation of peroxules and stromules (Supplementary Fig. 5)[25,26].

Autophagy is required for the degradation of damaged and toxic materials generated by ROS accumulation during oxidative stress[13]. However, the primary origin of ROS in leaf mesophyll cells of the autophagy-deficient mutants under photosynthetic conditions remains unclear. We hypothesised that the undegraded peroxisomes would primarily produce ROS in mutants during photorespiration-associated metabolism. A previous study showed that hydrogen peroxide accumulation is higher in peroxisomes than in chloroplasts and mitochondria during photorespiration[37]. Furthermore, the H$_2$-DCF-stained aggregates of peroxisomes in the mutants confirmed the accumulation of ROS in degrading peroxisomes (Fig. 7a, b and Supplementary Fig. 30c–e). Hydrogen peroxide in peroxisomes is immediately degraded by catalase in wild-type plants; however, catalase is gradually inactivated by increasing levels of ROS in photosynthetic tissues under high-intensity light conditions[5,38,39]. The inactivation of catalase causes over-accumulation of ROS in peroxisomes and induces the imbalance in ROS homeostasis within cells, leading to damage and defective plant growth in mutants[5,9,10]. The overexpression of catalase suppressed the increase in peroxisome numbers and aggregation in _atg2_ leaf cells (Supplementary Fig. 23); thus, suggesting that active catalase would reduce peroxisomal ROS and subsequent peroxisome aggregation in _atg2_. Peroxisomes participate in photorespiration through physical interaction with chloroplasts and mitochondria[40]. Therefore, damaged peroxisomes with high ROS levels should be immediately removed via pexophagy to maintain efficient metabolite flow among these organelles during photorespiration under high-intensity light conditions.

We focused on ATG18a, which is involved in the degradation of oxidised proteins[27], to assess the mechanism through which autophagy degrades peroxisomes. We used the ATG18a-GFP form because it complements the mutant phenotype of _atg18a(p2)_ (Supplementary

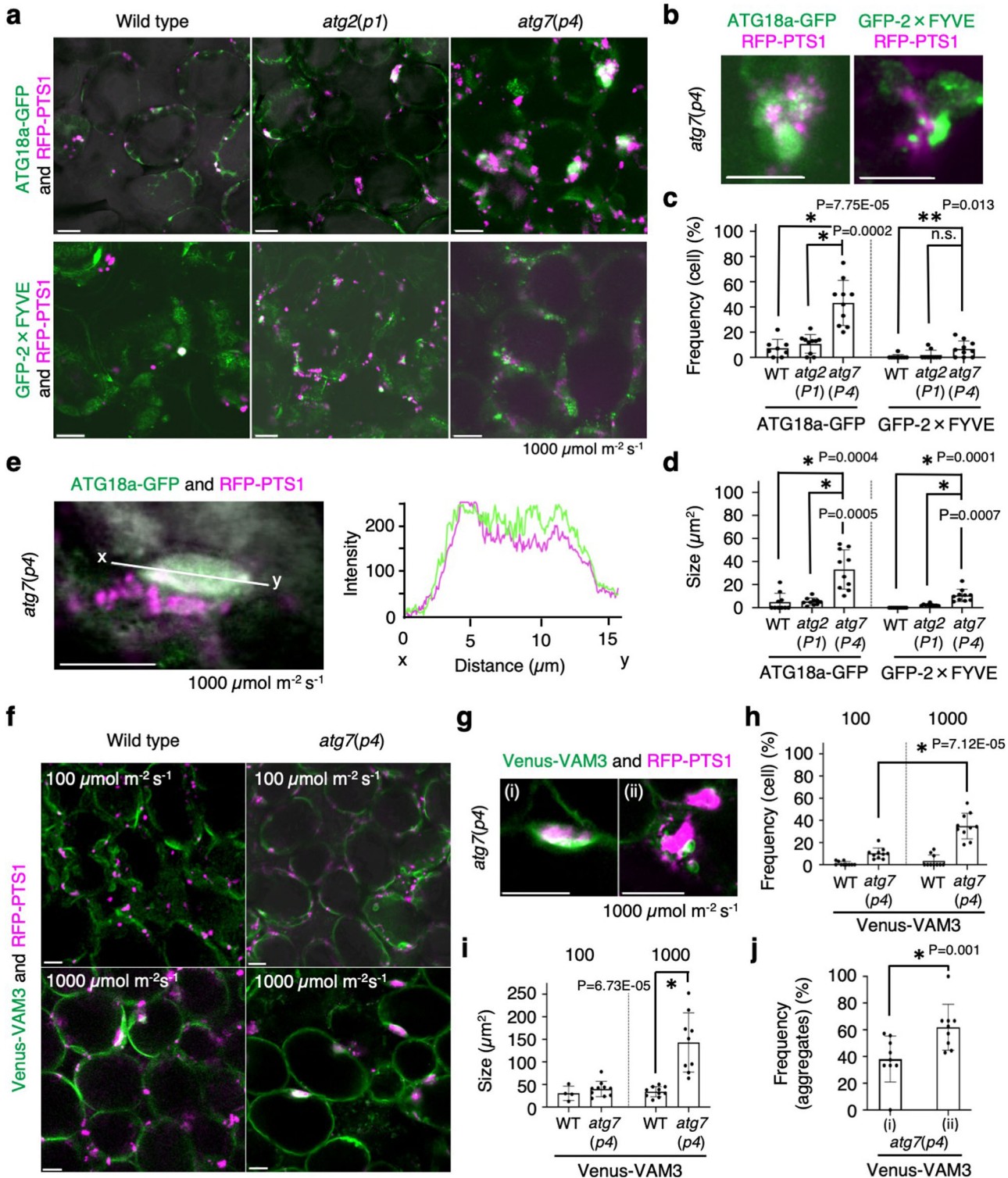

Fig. 18c, d), indicating that the fusion protein is functional. Since ATG18a has a well-conserved PtdIns3P-binding domain in yeast, plant, and animals[21,22,41,42], we used GFP-2×FYVE to monitor cellular PtdIns3P. Both GFP-2×FYVE and ATG18a-GFP preferred to target peroxisomal aggregates in *atg2(p1)* and *atg7(p4)* under normal light conditions (100 µmol m⁻² s⁻¹; Figs. 2, 3). Furthermore, we showed that exposure to high-intensity light (1000 µmol m⁻² s⁻¹) increased the frequency and size of peroxisome aggregates in *atg2(p1)*, *atg5*, and *atg7(p4)* mutants (Fig. 4b–d and Supplementary Figs. 21a, b, 22a, b), with an increase in GFP-2×FYVE and ATG18a-GFP targeting (Fig. 5c and Supplementary

Table 6). These proteins form autophagosome-like cup and ring structures that surround peroxisomes. The peroxisome number was increased in the vacuole of ConA-treated wild-type cells in the high-intensity light compared to the low-intensity light (Supplementary Fig. 24a–c). The luminal accumulation of peroxisomal RFP-PTS1 was observed in isolated vacuoles from high-intensity light-treated wild-type cells, but not the *atg7(p4)* cells (Fig. 6a–f and Supplementary Figs. 24a, d, e, 27a–d, e, 28a–f, 29a–c, d). These findings indicate that the light-induced peroxisome aggregates are specifically degraded in the vacuole via pexophagy. The peroxisomal aggregates in *atg2(p1)*

**Fig. 5 | Large aggregates of peroxisomes are surrounded by ATG18a-GFP and vacuolar membranes. a** Merged images of peroxisomes (RFP-PTS1, magenta) and ATG18a-GFP (green, upper row) or GFP-2×FYVE (green, lower row) in WT, *atg2(p1)*, and *atg7(p4)* after adaptation to high-intensity light (1000 μmol m$^{-2}$ s$^{-1}$) produced by an LED equipment (blue and red light) for 16 h. Plants were grown in the same condition as that shown in Fig. 4. **b** Enlarged images of peroxisome aggregates targeted by ATG18a-GFP and GFP-2×FYVE in *atg7(p4)* cells. **c** Frequency of cells containing large aggregates of peroxisomes surrounded by ATG18a-GFP (left) or GFP-2×FYVE (right). More than 135 cells were tested in each line. **d** Size of the large aggregates of peroxisomes surrounded by ATG18a-GFP (left) or GFP-2×FYVE (right). Number of structures examined: WT (*n* = 8), *atg2(p1)* (*n* = 15), and *atg7(p4)* (*n* = 58) for ATG18a-GFP; WT (*n* = 1), *atg2(p1)* (*n* = 2), and *atg7(p4)* (*n* = 10) for GFP-2×FYVE. **e** Plot profile of line *x*−*y* on a peroxisome aggregate. Magenta and green lines indicate the signals from RFP-PTS1 and ATG18a-GFP, respectively. **f** Fluorescence images of the vacuolar membrane (Venus-VAM3, green) and the large aggregate of peroxisomes (RFP-PTS1, magenta) in WT and *atg7(p4)*. **g** The large peroxisome aggregates surrounded (i) or not surrounded (ii) by the depressed region of the vacuolar membrane (i.e., vacuolar cavity) in *atg7(p4)* under high-intensity light conditions. **h** Frequency of cells with peroxisome aggregates surrounded by vacuolar membranes in WT and *atg7(p4)* plants. More than 200 cells were tested. **i** Size of the vacuolar-membrane structures surrounding the large aggregate of peroxisomes. The numbers of the structures were 11 (WT, high-intensity light) and 66 (*atg7(p4)*, high-intensity light). **j** Frequency of peroxisome aggregates surrounded (i) or not surrounded (ii) by the vacuolar membrane against the total number of peroxisome aggregates in *atg7(p4)* in **g**. The error bars indicate mean ± standard deviation (at least 8 biologically independent replicates), and asterisks indicate significant differences between WT and *atg2(p1)* or *atg7(p4)* (**P* < 0.01, two-sided Student's *t*-test) in (**c**, **d**, **h**–**j**). Scale bars in (**a**, **b**, **e**–**g**), 10 μm. The representative images in (**a**, **b**) and (**e**–**g**) show a summary of at least five independent experiments.

and *atg7(p4)* consist of oxidative peroxisomes with inactive catalase[11], and are recognised by both ATG8 and ATG18a (Supplementary Figs. 6, 7). Therefore, ATG18a recognises the oxidative peroxisomes by binding with PtdIns3P to degrade them. We further showed that overexpression of ATG18a-GFP suppresses ROS in both cells and peroxisomes under high-intensity light conditions (Supplementary Fig. 32), providing supporting evidence for the contribution of ATG18a in light-dependent pexophagy meant to prevent cell damage.

ATG18a-GFP was occasionally localised to places other than peroxisomes (Fig. 2d), such as chloroplasts (Supplementary Fig. 31) and undefined structures in the cell (Fig. 2a, d, and Supplementary Fig. 9a and Supplementary Table 3), suggesting that some of the chloroplasts and other cellular materials are degraded by autophagy under the high-intensity light conditions. In our tested-light conditions, ATG18a-GFP and GFP-2×FYVE recognised chloroplasts with lower chlorophyll fluorescence, presumably chloroplasts damaged following exposure to high-intensity light (Supplementary Fig. 31). This is consistent with previous reports showing that high-intensity light induces ROS accumulation in chloroplasts[26] and subsequent degradation of damaged chloroplasts by autophagy (chlorophagy)[43]. Meanwhile, the relative intensity of $H_2$-DCF in peroxisomes in *atg2(p1)* and *atg7(p4)* was approximately three times stronger than that in chloroplasts (Supplementary Fig. 30c, f). We also discovered that autophagy had a slight contribution to the degradation of mitochondria (mitophagy) but to a lesser extent than pexophagy (Supplementary Fig. 25). We noticed that HSP70s were recovered in the pull-down assay of ATG18a-GFP (Supplementary Fig. 13 and Supplementary Table 2), implying the involvement of chaperone-mediated autophagy or microautophagy[44]. Collectively, these findings suggest that various types of cellular components, mostly damaged peroxisomes, are degraded via autophagy under high-intensity light conditions.

Selective autophagy has been well studied in yeast[15,18,22] and mammals[23,45,46] but less in plants[13,17]. The subcellular location of PtdIns3P synthesis during autophagy differs depending on the organisms and organelles to be degraded (e.g., PAS in yeast and omegasomes in mammals)[46–50]. In plants, the location of PtdIns3P synthesis, the origin of isolation membranes, and the mechanism through which ATGs participate in pexophagosome formation and degradation are unknown[13,17,51]. We showed that numerous dot structures of ATG18-GFP (Fig. 2a, e, f) and GFP-2×FYVE (Fig. 3a, e, f) localised to peroxisomes in *atg2(p1)* and *atg7(p4)*, suggesting that PtdIns3P is formed adjacent to the peroxisomes to attract ATG18a before the action of ATG2 and ATG7 in the process of macropexophagy. Detailed analysis by electron microscopy revealed that PtdIns3P and ATG18a were localised on both peroxisomes and phagophores adjacent to peroxisomes in *atg2(p1)* (Supplementary Figs. 15b, 16), similar to ATG8 (Supplementary Fig. 17).

Recent studies have shown that phagophores in mammalian cells are generated from the contact site between the ER and mitochondria[41,42,45,47,52] and from the ER in which ATG5, ATG9, and ATG18 are localised in plant cells[53,54]. In yeast *Saccharomyces cerevisiae*, ATG2–ATG18 complex tethers PAS to the ER for extending the isolation membrane[55]. We showed that the ER and phagophores were located adjacent to the high-density area in peroxisomes of *atg2(p1)* (Supplementary Figs. 15b, c, 16, 17) and *atg5* mutants[14]. PtdIns3P, ATG18a, and ATG8 were localised to the same area (Figs. 2, 3 and Supplementary Figs. 15b, 16, 17). These findings suggest that the initial phagophore generates at the site where the ER overlaps with a specific receptor and the PtdIns3P on peroxisomes in plant macropexophagy, acting as a platform for PAS. ATG18 gathers at the PtdIns3P on the membrane for pexophagosomes extension with a lateral supply of the isolation membrane from the ER. Disturbance of the subcellular localisation of the ATG18a-GFP and GFP-2×FYVE on degraded peroxisomes in wortmannin-treated *atg2(p1)*, *atg7(p4)*, or *atg18a(p2)* (Supplementary Fig. 19) implies that phosphoinositide 3-kinase activity is required to form the platform for gathering ATG18a on degraded peroxisomes in the macropexophagy process.

After initiation, the phagophore elongates to cover the peroxisome and becomes a pexophagosome, which then enters into vacuoles for degradation. Lack of autophagy causes the accumulation of damaged peroxisomes and consequently leads to peroxisome aggregation. In wild-type and *atg7(p4)* cells, we observed that the dot structures of ATG18-GFP or GFP-2×FYVE gradually change to ring structures with a cup structure to engulf peroxisomes; however, this change was not observed in *atg2(p1)* (Supplementary Figs. 10, 14 and Supplementary Movies 5–7, 10, 11). This indicates that ATG2 and ATG18a play an indispensable role in enveloping the degraded peroxisomes with ATG8-PE to form pexophagosomes and induce macropexophagy. The aggregated peroxisomes were captured by invagination into vacuoles in *atg7(p4)* (Fig. 5f, g, and Supplementary Fig. 26). This was also seen in isolated vacuoles (Fig. 6 and Supplementary Figs. 27). ATG7 plays a role in the maturation of ATG8-PE as a ubiquitin-activating enzyme-like protein for generating autophagosomes[56,57]. We have previously revealed that ATG8a localises to degraded peroxisomes in *atg2(p1)* and *atg5* as dot structures[11,14]. In this study, we showed that ATG8e co-localises with ATG18a on peroxisome aggregates in *atg2(p1)* and atg7(*p4*) (Supplementary Figs. 6 and 7); thus, suggesting that ATG8 acts in concert with ATG18a. Leaf damage and peroxisome aggregation in *atg9* are reduced compared to those in *atg2*, *atg5*, and *atg7* under high-intensity (Supplementary Figs. 20–22) and normal light conditions[11,14], suggesting that the contribution of ATG9 in plant macropexophagy is reduced, unlike in yeast and mammal pexophagy[18,19,49,50]. ATG9 might not have a specific role in pexophagy, although it is generally required for autophagy in plants.

We discovered that the vacuolar membrane forms large cavities to surround peroxisome aggregates in *atg7(p4)* under high-intensity light conditions (Fig. 5f–j and Supplementary Fig. 26). Furthermore, some of the peroxisomes and peroxisome aggregates on the surface of the

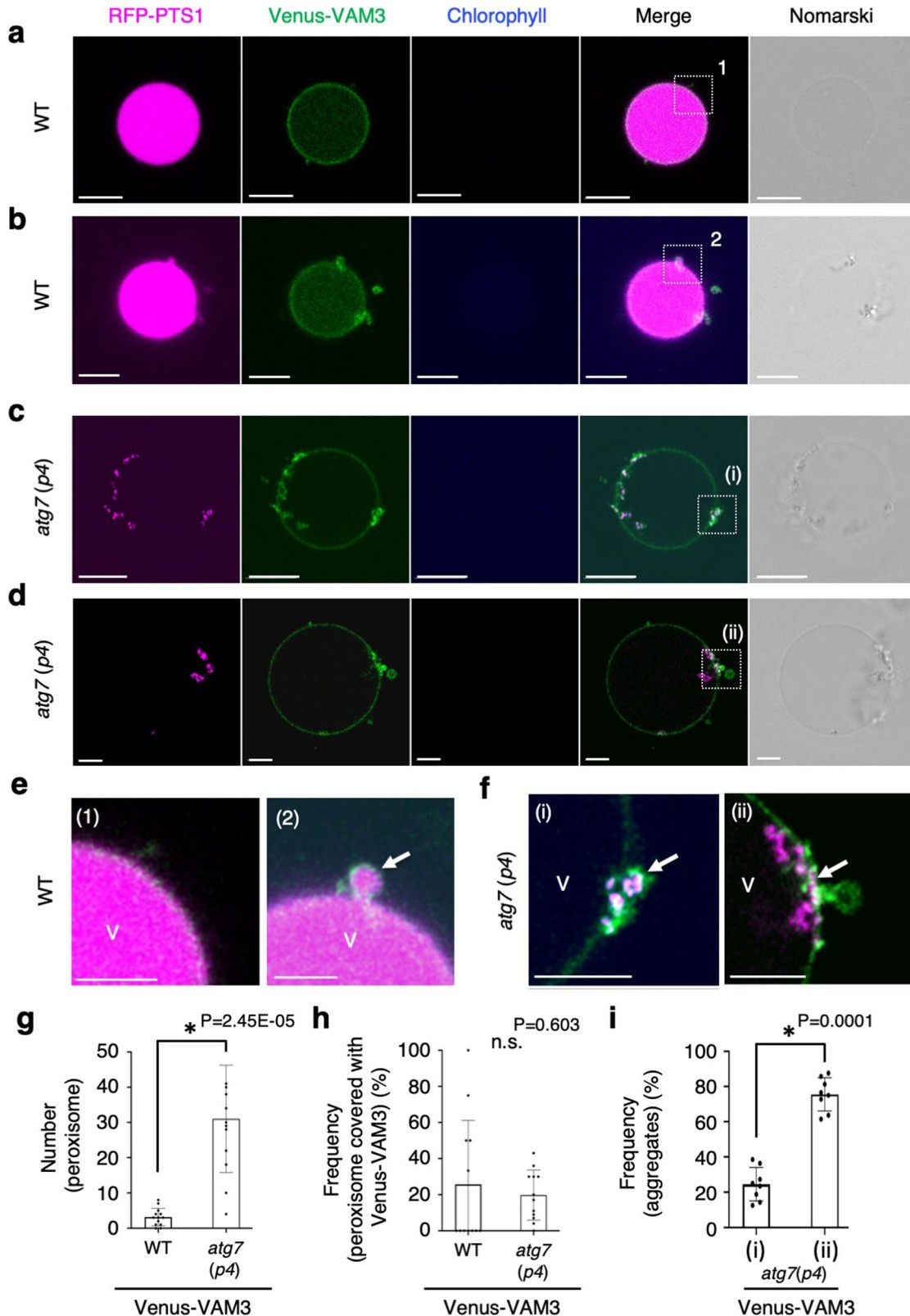

isolated vacuole in wild-type and *atg7(p4)* cells were also surrounded by the vacuolar membrane (Fig. 6 and Supplementary Figs. 27 and Supplementary Movies. 18–21). These direct actions of the vacuolar membrane indicate the involvement of the microautophagy process during the incorporation of degraded peroxisomes into the vacuole. In plants, microautophagy is induced in sucrose-starved root cells[58]. Microautphagy contributes to the accumulation of anthocyanin

aggregate in vacuoles[59] and damaged chloroplast degradation under high-intensity light irradiation[60,61]. Microautophagic degradation of peroxisomes (micropexophagy) was reported in yeast, where it is accompanied by a micropexophagy-specific apparatus (MIPA)[22], but not yet in plants. Taken together, these findings suggest that micropexophagy is induced following high-intensity light exposure resulting in the degradation of oxidized peroxisomes and their aggregates.

**Fig. 6 | Large aggregates of peroxisomes are accumulated on the surface of isolated vacuoles. a–d** Images of peroxisomal and vacuolar RFP-PTS1 (magenta), Venus-VAM3 (green), chlorophyll (blue), merge, and Nomarski in isolated vacuoles from WT (**a**, **b**) and *atg7(p4)* (**c**, **d**) plants after high-intensity light (1000 μmol m⁻² s⁻¹) treatment with an LED equipment (blue and red light) for 5 h. Scale bars in (**a–d**), 5 μm. Plants were grown in the same condition as shown in Fig. 5. **e, f** Enlarged images of isolated vacuoles (V) with peroxisomes in WT (**e**) and peroxisome aggregates in *atg7(p4)* (**f**) in white-dots line squares (**a–d**). Scale bars in (**e**, **f**), 2 μm. White arrows indicate peroxisome (**e** (2)) and peroxisome aggregate (**f** (i),(ii)). **g** Number of

peroxisomes in isolated vacuoles from WT and *atg7(p4)* cells. **h** Frequency of peroxisomes surrounded by the vacuolar membrane. (*n* = 12 biologically independent vacuoles). **i** Frequency of peroxisome aggregates surrounded (i) or not surrounded (ii) by the vacuolar membrane on the surface of the isolated vacuole in **g** (*n* = 8 biologically independent vacuoles with peroxisome aggregates). The error bars indicate mean ± standard deviation, and asterisks indicate significant differences between WT and *atg7(p4)* (*P* < 0.01, two-sided Student's *t*-test) in (**g**, **h**). The representative images in (**a–f**) show a summary of at least eight independent vacuoles from three independent experiments.

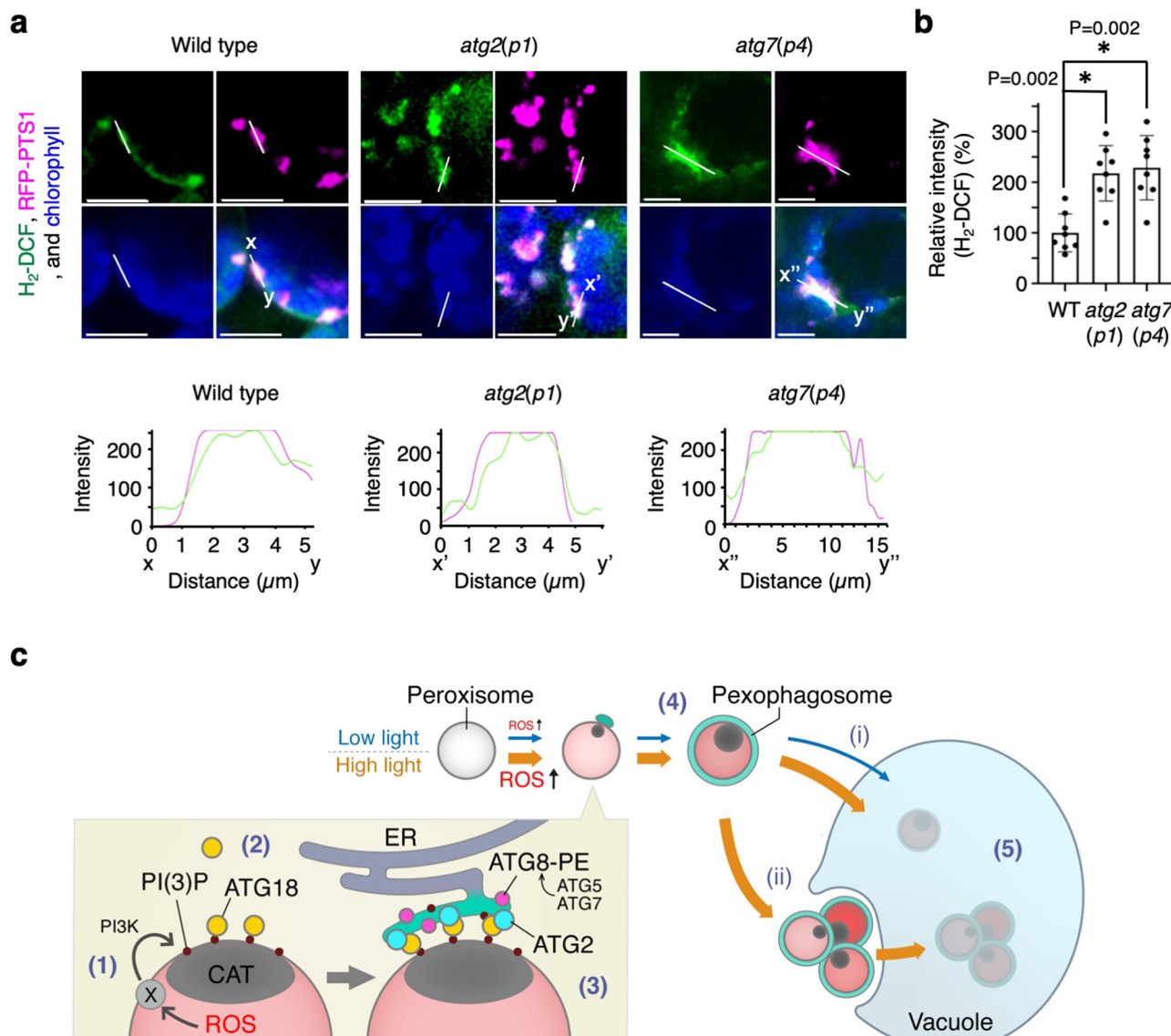

**Fig. 7 | ROS accumulation in peroxisomes and a model of pexophagy under high-intensity light conditions. a** Detection of ROS using H₂-DCF staining (green) after exposure to high-intensity light of WT, *atg2(p1)*, and *atg7(p4)* plants. Green, magenta and blue indicate H₂-DCF, peroxisomes (RFP-PTS1), and chloroplasts (autofluorescence), respectively. Plants were grown and adapted to high-intensity light similar to that shown in Figs. 4, 5. Plot profile of fluorescence intensity on the lines; *x–y* (WT), *x′–y′* (*atg2(p1)*), and *x″–y″* (*atg7(p4)*) depicted in each upper image. Scale bars, 10 μm. **b** Relative fluorescence intensity of H₂-DCF in peroxisome aggregates in WT, *atg2(p1)*, and *atg7(p4)*. A total of 140 peroxisomes from each line were tested. The error bars indicate mean ± standard deviation (*n* = 8 biologically independent replicates), and asterisks indicate significant differences between WT and *atg2(p1)* or *atg7(p4)* (*P* < 0.01, two-sided Student's *t*-test). The representative

images in **a** show a summary of at least five independent experiments. **c** Model for ROS-induced peroxisome degradation via pexophagy. (1) During photorespiration under light, peroxisome gradually accumulates ROS, which impairs the catalase function (dark grey). PtdIns3P is generated on the peroxisome membrane or associating PAS via the action of an undefined factor (Factor X) in response to ROS. (2) PtdIns3P on/around the peroxisomal membranes are recognised by ATG18a, which proceeds the formation of pexophagosomes close to the ER. (3) This process occurs in cooperation with ATG2 and ATG8-PE, which are modified by other ATGs such as ATG5 and ATG7. (4) The damaged peroxisome is enveloped by the pexophagosome, (5) and is subsequently degraded in the vacuole via macropexophagy (i) and micropexophagy (ii).

Since the microautophagic degradation of peroxisome aggregate seems incomplete in *atg7(p4)*, ATG7 and ATG8-PE are probably required for microautophagy.

Collectively, our data suggest that ATG2, ATG5, ATG7, ATG8, and ATG18a work cooperatively to generate complete pexophagosomes and degrade them in vacuoles via macro- and micropexophagy. Based on these results, we propose the following model for macropexophagy (Fig. 7c): (1) peroxisomes with inactive catalase accumulate high levels of ROS, and PtdIns3P is generated on the peroxisome membrane or phagophores formed adjacent to peroxisome and ER; (2) ATG18a targets the PtdIns3P on the damaged peroxisomes; (3) pexophagosomes are formed by ATG18a and PtdIns3P along with other autophagy factors; (4) pexophagosomes completely sequester damaged peroxisomes; (5) pexophagosomes are incorporated into the vacuole. We provided the scheme of the process of degradation and formation of the peroxisome aggregates and their degradation in wild type, *atg2(p1)*, and *atg7(p4)* via macro- and micropexophagy (Supplementary Fig. 35). Under normal-intensity light conditions, the *atg7(p4)* mutant showed a higher number of peroxisome aggregates compared to the other *atg* mutants (Fig. 1 and Supplementary Fig. 1). The difference may reflect the function of ATG2 and ATG7 protein in macropexophagosome formation (Figs. 2, 7c and Supplementary Figs. 9, 35). Conversely, no difference was observed in the degree of peroxisome aggregation under high-intensity light conditions between *atg2* and *atg7* (Supplementary Figs. 20–22); thus, suggesting that ATG2 and ATG7 play equally important roles in micropexophagy.

We speculate that ROS generation is responsible for the induction of pexophagy; however, it is yet unclear how ROS generated in the peroxisome matrix are recognised for pexophagy. In human pexophagy, ataxia-telangiectasia mutated protein on the peroxisomal membrane senses ROS inside the peroxisome to induce pexophagy by mediating mTORC1 suppression and peroxin 5 (PEX5) phosphorylation[12,62]. Plant pexophagy might also involve sensor protein(s) along with plant PEX proteins on the peroxisome membrane to induce pexophagy[12,22,62]. In yeasts, receptors such as PpAtg30 and ScAtg36 interact with PEX3 and PEX14 to recognise peroxisomes to be degraded in pexophagy; however, orthologues of these receptors were not identified in plants[12,22,23,49,51,62,63]. Alternatively, oxidised lipids on the peroxisome membrane may represent the signal to induce pexophagosome formation as they are the hallmark of oxidised peroxisomes. PtdIns3P accumulation takes place in both peroxisomes and phagophores. This is supported by the fact that multiple pathways for the accumulation of PtdIns3P are activated in autophagy[46,48,64]. During mitophagy in mammalian cells, activation of phosphoinositide 3-kinase and inactivation of PTEN, a phosphatase removing the phosphate in the D3 position of the inositol ring, occur on the membrane of initial phagophores, namely omegasomes[41,42,45,47], which are derived from the ER as platforms for mitophagy[65,66]. Here, we showed that phosphoinositide 3-kinase is involved in pexophagosome formation (Supplementary Fig. 19). The future direction of this study is to find the ROS or oxidative lipid sensor protein(s) on the peroxisome membrane for activating the phosphoinositide 3-kinase to induce pexophagy and clarify the involvement of PEXs in plant pexophagy.

We demonstrated that ATG18a-GFP selectively targets and surrounds peroxisomes to be degraded, which is the first observation of pexophagosomes forming from phagophores in plant cells. Hence, our analysis provides deep insight into the mechanism underlying autophagosome formation. Furthermore, our findings allow a better understanding of how plants reduce ROS production via autophagy to improve photosynthetic efficiency and thus increase crop yield.

## Methods

### Plant material and growth conditions
Wild-type and transgenic plants were grown in a 16 h light/8 h dark cycle at 23 °C in an incubator (MLR-351, Sanyo Electric Co., Ltd.,

Japan)[11,14]. *Arabidopsis thaliana* (L.) Heynh (Columbia, Col-0) and that expressing *GFP-PTS1* (the GFP-PTS1 plant) or *RFP-PTS1* (the RFP-PTS1 plant)[67] were used as controls. The *atg2(p1)*, *atg18a(p2)*, and *atg7(p4)* (*peups*) plants were previously screened as pexophagy mutants[11], and T-DNA insertion lines *atg2-1* (SALK_076727), *atg5-1* (SAIL_129B07), *atg7-2* (GABI_655B06), and *atg9-3* (SALK_130796)[14] were used for plant growth analysis. The *peups* expressing *RFP-PTS1* (RP) were generated from F3 lines by crossing *peups* with the RFP-PTS1 plants. We produced the RFP-PTS1 plants expressing *ATG18a-GFP* or *GFP-2×FYVE* using the floral dip method[68] with *Agrobacterium tumefaciens* (EHA101) harbouring the binary vector pGWB451-ATG18a or pGWB452-2×FYVE. More than three independent lines that showed normal growth phenotypes similar to Col-0 were selected (Supplementary Fig. 18c, d). The *peups* (RP) expressing *ATG18a-GFP* or *GFP-2×FYVE* were generated by crossing RFP-PTS1 plants expressing *ATG18a-GFP* or *GFP-2×FYVE* with *peups* (RP). These lines theoretically express the transgenes at the same level. Transgenic *Arabidopsis* expressing *Venus-VAM3* (vacuolar membrane marker)[58,69] or *Mt-GFP* (mitochondrial marker)[70] were crossed with the RFP-PTS1 plant and *peups* (RP), respectively, to generate T3 homozygous lines.

### Plant growth analysis under high-intensity light conditions
One week after germination on 0.8 % (w/v) agar plates containing half-strength MS medium and 1% (w/v) sucrose at 23 °C in a 16 h light (50 $\mu$mol m$^{-2}$ s$^{-1}$)/8 h dark photoperiod, plants were transferred to rockwool inserted into the soil under 50 $\mu$mol m$^{-2}$ s$^{-1}$ white light (OSRAM FL25W White, Hitachi, Japan) at the same photoperiod for 2 weeks and then placed in incubators with white light at 50, 100, and 200 $\mu$mol m$^{-2}$ s$^{-1}$ for plant growth analysis (Supplementary Fig. 2). After the plants were grown at 23 °C in the 16 h light (normal white light, 100 $\mu$mol m$^{-2}$ s$^{-1}$)/8 h dark cycle for 3 weeks on 0.8% (w/v) agar containing 1% (w/v) sucrose and 1× MS salt, plant samples were used for the plant growth and biochemical analysis. The plant growth analysis (Figs. 4–7 and Supplementary Figs. 20–34) involved irradiation with blue (450 nm) and red (640 nm) light using an LED equipment (ISC-150 × 150-H4RB45; CCS, Japan) with a power supply (ISC-201-2; CCS, Japan) in low and high-intensity light conditions at 100 and 1000 $\mu$mol m$^{-2}$ s$^{-1}$ for 16 h, respectively.

### Vector construction
Binary vectors pGWB451-ATG18a-G3-GFP and pGWB452-G3-GFP-2×FYVE were constructed using the Gateway system (Thermo Fisher Scientific, Waltham, MA, USA) to transform RFP-PTS1 plants. Adapter-tagged cDNA of AtATG18a (At3g62770, accession no. NM_116142) was generated by PCR by amplifying the corresponding region using the following primer set: F: 5′-TACAAAAAAGCAGGCTTCATGGCCACCGTA TCTTCTTC-3′, R: 5′-GTACAAGAAAGCTGGGTTGAAAACTGAAGGCGGT TTCAGA-3′ for ATG18a, which was then recombined with the pDONR™221 vector[71].

Adapter-tagged cDNA of the FYVE domain[32,72] was generated using PCR by amplifying the corresponding region using the following primer set: *attB*1-adapter, 5′-GGGGACAAGTTTGTACAAAAAAGCAG GCT TC-3′; and *attB*2-adapter, 5′-GGGGACCACTTTGTACAAGAAAGCTGGG TT-3′ using the pBluescript KS (-) (Stratagene) vector containing the nucleotide sequence of the 2×FYVE domain with *attB*1 and *attB*2 as templates and then recombining with the pDONR™221 vector. The nucleotide sequences of two FYVE domains, *attB*1-FYVE and FYVE-*attB*2, were separately inserted in the same pBluescript KS (-) vector in two steps using the FYVE region of *Mus musculus* HGF-regulated tyrosine kinase substrate (Hgs; accession no.: NM_001159328) as the cDNA template[32,72]. Two primer sets, F1: 5′-AAGTCGACTACAAAAAAG CAGGCYYCGAAA GTGATGCCATGTTCGCTG-3′ and R1: 5′-AAAAGCTT GACCTTGTGCCTTCTTGTTCAGCTGCTCATA-3′ for *attB*1-FYVE, and F2: 5′-AAAAAGCTTCTGAAAGTGATGCCATGTTCGCTGCTGAAA-3′ and R2: 5′-AAGATTCGTACAAGAAAGCTGGGTGCCTTCTTGTTCAGCTGCT

CATA-3′ for FYVE-*attB*2 were used to amplify the corresponding regions.

## Imaging analysis

A confocal laser scanning microscope (LSM 510, LSM880, Zeiss, Germany) with a ×40 or ×63 objective was used for imaging analyses of peroxisomes and for determining the intracellular distribution of fluorescent proteins as described previously[11,40]. Images were obtained from the top surface to the middle depth region, "Top", or the middle depth to the bottom region, "Bottom", of a 3-week-old plant leaf. We used a slice from the *z*-axis scanning image taken at every 1 μm thickness with a pinhole size of 1 Airy unit. The excitation and emission wavelengths for the images were 488 and 492−570 nm, respectively, for GFP, and 516 and 600−625 nm, respectively, for RFP. Time-lapse images were obtained for 250−300 s with a temporal resolution of 5 s, and movie files were generated using Fiji (ImageJ, NIH public domain). The number of cells and organelles was counted using the Analyze Particles and Cell Counter plugins equipped in Fiji[73]. The size of peroxisome aggregates was measured manually using the polygon selection tool in Fiji after the images were magnified three-fold for precise selection of the periphery. The pexophagosome around peroxisomes in *atg7*(*p4*) targeted by ATG18a-GFP (Fig. 2h) was identified by conducting mathematical morphology analyses[28] based on time-lapse images. Fluorescence intensity (Figs. 2g, 3g, 5e, 7a, and Supplementary Fig. 31b) was measured using the Plot Profile plugin equipped in Fiji. FRAP analysis (Supplementary Figs. 11, 12) was performed using LSM880 with an Ar laser (488 nm) at 50% intensity to induce photobleaching. Images were obtained every 1 s, and fluorescence intensity was then measured using Fiji.

## Measurement of chlorophyll content and photosynthetic efficiency

Chlorophyll content (Supplementary Figs. 2b, 20b, 32b) was measured as previously described[74] using the rosette leaves adapted to each light intensity. Photosynthetic efficiency (Supplementary Fig. 2c) was measured as the maximum yield of photosynthesis system II using a photosynthesis yield analyser (MINI-PAM; Walz, Effeltrich, Germany)[75] using at least three leaves from five plants after they were adapted to each light intensity. Three independent experiments were performed.

## Electron microscopy analysis

Electron microscopy analysis was performed following previous works[11,76]. Three-week-old wild-type and *atg7*(*p4*) plants were analysed for catalase accumulation (Supplementary Figs. 3, 15c), chloroplast and peroxisome membranes (Supplementary Fig. 5e), and mitochondria (Supplementary Fig. 25b). Plant leaves were fixed in 4% (w/v) paraformaldehyde, 1% (w/v) glutaraldehyde, and 0.06 M sucrose in 0.05 M cacodylate buffer (pH 7.4) for immunoelectron microscopy analyses with antibodies against peroxisomal proteins [malate synthase (MS), isocitrate lyase (ICL), glycolate oxidase (GO), hydroxypyruvate reductase (HPR), and catalase (CAT)] (Supplementary Figs. 3, 15c)[11,76]. Three-days-old wild-type and *atg2*(*p1*) plants expressing GFP-2×FYVE and ATG18a-GFP were analyzed to detect the locations of PtdIns3P and ATG18 (Supplementary Figs. 15b, 16, 17)[11,76]. For immunoelectron microscopy analyses with antibodies against GFP, cotyledons were frozen with a high-pressure freezing machine (HPM-010, Bal-Tec, Balzer, FL) and dehydrated by freeze-substitution methods. Samples were embedded in LR white. Sections were treated with anti-GFP antibodies (1:100−1000 (v/v)) for 1 h at room temperature, and then treated with protein A-gold (15 nm, BBI international) for 30 min. Sections were stained with 4% (w/v) uranyl acetate for 10 min at room temperature and examined under a transmission electron microscope (H-7650, Hitachi High-Tech Co.) at 80 kV[11,76].

## NBT and H₂-DCF staining

Nitro blue tetrazolium (NBT) and 27′-dichlorodihydrofluorescein ($H_2$-DCF) staining were performed as follows: rosette leaves of GFP-PTS1, *atg2*(*p1*), and *atg7*(*p4*) were immediately submerged in NBT (Sigma-Aldrich, St. Louis, MO, USA) solution for 1 h, and chlorophyll was then repeatedly removed with 100% ethanol in 95 °C water for 10 min and washed with pure water. In the case of $H_2$-DCF staining, the leaves were submerged in 10 μM $H_2$-DCF-DA (Thermo Fisher Scientific) for 10 min and then washed once with pure water[33,34]. At least three independent experiments were performed. We carefully selected leaf mesophyll cells from similar regions and depths within the leaves and used the same confocal microscope setting of exposure time and dynamic range across images. The mean intensity of fluorescence from $H_2$-DCF inside peroxisome and chloroplast was measured using Image J. The area of peroxisome and chloroplast was determined by surrounding them with the "Polygon selections tool" in Image J.

## Immunoblot analysis

Immunoblotting was performed following a previous work[11]. Total proteins of wild-type, *peups*, *atgs*, and various transgenic plants grown under different light intensities for 1−2 days were extracted with the extraction buffer containing 10 mM HEPES−KOH (pH 8.0) and a protease inhibitor cocktail (Roche). Total proteins were then fractionated into supernatant and pellet by centrifugation at 20,000×*g* for 10 min at 4 °C. The pellet was washed with extraction buffer twice, followed by solubilisation with extraction buffer containing 1% (w/v) SDS. Each 10 μg of total protein was separated by SDS−PAGE and transferred onto a polyvinylidene difluoride membrane (Millipore, Billerica, MA, USA) in a semidry electroblotting system (BioCraft). Immunoblot analyses were subsequently performed using antibodies against peroxisomal proteins CAT, peroxin 14 (PEX14), GO, ascorbate peroxidase (APX), and HPR[11], as well as against mitochondrial proteins cytochrome c oxidase 2 (COXII) (Agrisera, Sweden) and serine hydroxymethyltransferase (SHMT) (Agrisera, Sweden). We captured immunoblot images with a CCD camera by using precisely the same parameters for all conditions (exposure time, contrast, and background intensity). Signal intensities of bands in the immunoblot image were quantified using Dot Blot Analysis in Fiji. The CAT, COXII, and SHMT amounts in the supernatant and pellet fractions were calculated using volume-based normalisation of the extraction buffer (Source Data: Fig.4 and Supplementary Figs. 22, 25).

## Mass spectrum analysis

Total protein was extracted from *atg2*(*p1*) expressing *ATG18a-GFP* or *GFP* grown under normal light conditions with 1 mL of lysis buffer [50 mM HEPES−KOH (pH 7.5), 0.15 M NaCl, 0.5% (v/v) Triton X-100, and 0.1% (v/v) Tween 20]. ATG18a-GFP-binding proteins were obtained through immunoprecipitation using μMACS Anti-GFP MicroBeads and μMACS columns (Miltenyi Biotec, Gaithersburg, MD, USA)[77]. The eluted fraction was assessed by immunoblot analysis using an anti-GFP antibody to detect GFP or ATG18a-GFP (Supplementary Fig. 13a). ATG18a-GFP/GFP-binding proteins were subjected to SDS−PAGE following in-gel digestion[77]. The obtained proteins were electrophoresed briefly until the BPB dye band was 2 mm from the well. A 4-mm piece of gel centred on the dye band was cut out and digested with trypsin[78,79]. Collected peptides were analysed using nano-LC−MS/MS (LTQ Orbitrap XL; Thermo Fisher Scientific)[78,79]. The obtained spectra were searched against the TAIR 10 Arabidopsis protein database (version 20101214) with MASCOT server (version 2.3.02, Matrix Science, London, UK)[78,79]. The list of identified proteins is shown in Supplementary Table 2. The experiments were repeated three times.

## Lipid binding assay

The binding ability of ATG18a-GFP to PtdIns3P was determined using PIP Strips P-6001 (Echelon Biosciences Inc., Salt Lake City, UT, USA) according to the manufacturer's instructions and Tamura et al. (2013)[31]. Crude extracts from *atg2(p1)* expressing *ATG18a-GFP* and wild type (Col-0) expressing GFP were incubated with PIP strips for 3 h at 23 °C after removing debris by centrifugation at 1000×*g* for 5 min. After two washes with TBS containing 0.1% (v/v) Tween 20, the binding of ATG18a-GFP to lipids was detected using an antibody against GFP and ImageQuant LAS4000 (GE Healthcare) at high sensitivity mode.

## Reporting summary

Further information on research design is available in the Nature Portfolio Reporting Summary linked to this article.

## Data availability

The source data for Figs. 1–7 and Supplementary Figs. 1–34 are provided with this paper as a Source Data file. Other data and materials of this study are available from the corresponding author upon reasonable request. Source data are provided with this paper. Proteome data were deposited in PRIDE with accession number (PXD038480).

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

## Acknowledgements

We thank Dr. Shunichi Takahashi (National Institute for Basic Biology) and Dr. Murray Badger (Australian National University) for the helpful discussion about photoinhibition and photosynthetic efficiency in pexophagy mutants. We also thank Dr. Tsuyoshi Nakagawa (University of Shimane) for providing the Gateway vectors pGWB451 and pGWB452, Dr. Shinichi Arimura (University of Tokyo), and Dr. Tomohiro Uemura (Ochanomizu University) for kindly providing the transgenic line expressing *Mt-GFP* and the plasmid hovering *VAM3* gene, and the Bioimaging Facility in National Institute for Basic Biology (NIBB) as well as the NIBB BioResource Center for technical support. This work was supported by a Grant-in-Aid for Scientific Research on Innovative Areas to M.N. (no. 22120007) from the Ministry of Education, Culture, Sports, Science and Technology (MEXT); by Grants-in-Aid for Scientific Research to Y.H. and K.O. (no. 17K07467), to M.N. (no. 20370024), to S.M. (nos. 26440157 and 20570045), and to I.H.-N. (nos. 15H05776 and 22000014) from Japan Society for the Promotion of Science (JSPS); by the Japan Science and Technology Agency Exploratory Research for Advanced Technology program (JST-ERATO) to K.N. (no. JPMJER1602); by a SONATA-BIS Grant to S.G.-Y. (UMO-2019/34/E/NZ3/00299) from National Science Centre Poland; by a TEAM Grant to K.Ya. (TEAM/2017-4/41) from the Foundation for Polish Science; by the Wyeth Foundation to M.N. and I.H.-N.; and by the Hirao Taro Foundation of KONAN GAKUEN for Academic Research to I.H.-N. The open-access publication of this article was funded by the BioS Priority Research Area under the program "Excellence Initiative—Research University" at the Jagiellonian University in Krakow.

## Author contributions

K.O., M.S., K.Yo., Y.H., S.G.-Y., S.M., K.N., K.Ya., Y.O., and M.N. designed the study. K.O. performed most of the experiments. Y.H., M.K., and K.S. performed EM analysis. Y.K. performed mathematical morphology analyses. D.T. and M.U. performed protein mass spectrometry analyses. K.O., S.G.-Y., K.Yo., A.T., A.K., H.U., I.H.-N., K.H., and S.M. generated transgenic plants and performed plant growth analyses. All the authors analysed the data and wrote the manuscript. K.O., S.G.-Y., S.M., K.Ya., and M.N. mainly performed revised experiments and wrote revised manuscripts.

## Competing interests

The authors declare no competing interests.

## Additional information

[1]Department of Cell Biology, National Institute for Basic Biology, Okazaki 444-8585, Japan. [2]Malopolska Centre of Biotechnology, Jagiellonian University, Krakow 30-387, Poland. [3]Department of Science, Faculty of Science, Niigata University, Niigata 950-2181, Japan. [4]United Graduate School of Agricultural Sciences, Iwate University, Iwate 020-8550, Japan. [5]Department of Imaging Science, Center for Novel Science Initiatives, National Institutes of Natural Sciences, Okazaki 444-8787, Japan. [6]Laboratory of Biological Diversity, Department of Evolutionary and Biodiversity, National Institute for Basic Biology, Okazaki 444-8585, Japan. [7]RIKEN Center for Sustainable Resource Science, Yokohama 230-0045, Japan. [8]Department of Life Sciences, School of Agriculture, Meiji University, Kanagawa 214-8571, Japan. [9]Department of Biology, Graduate School of Sciences and Technology for Innovation, Yamaguchi University, Yamaguchi 753-8512, Japan. [10]Laboratory of Organelle Regulation, National Institute for Basic Biology, Okazaki 444-8585, Japan. [11]Faculty of Science and Engineering, Konan University, Kobe 658-8501, Japan. [12]Department of Plant-bioscience, Iwate University, Morioka 020-8550, Japan. [13]Biomacromolecules Research Team, RIKEN Center for Sustainable Resource Science, Wako 351-0198, Japan. [14]Cell Biology Center, Institute of Innovative Research, Tokyo Institute of Technology, Yokohama 226-8503, Japan. [15]Department of Basic Biology, School of Life Science, SOKENDAI (The Graduate University for Advanced Studies), Okazaki 444-8585, Japan. [16]Present address: Kyoto University, Katsura, Nishikyoku Kyoto 615-8510, Japan. [17]Present address: Graduate School of Science and Engineering, Saitama University, Saitama 338-8570, Japan. [18]Present address: Department of Management and Information Sciences, Faculty of Environmental and Information Sciences, Fukui University of Technology, Fukui 910-8505, Japan. ✉e-mail: kenji.yamada@uj.edu.pl; mramushini@gmail.com

