## [Peer Review File · Nature Communications]

Pexophagy suppresses ROS-induced damage in leaf cells under high-intensity lightReviewer #1 (Remarks to the Author):

In this manuscript, the authors take advantage of autophagy-defective (*atg*) mutants and various fluorescent reporters labeling peroxisomes, ATG18a, and PIP to examine pexophagy in Arabidopsis, especially in response to high-intensity light. The authors previously reported peroxisome aggregation in *atg* mutants, and here show that the pattern of aggregation differs in *atg2* and *atg7* mutants (Fig. 1). They show that ATG18a-GFP (Fig. 2) and GFP-FYVE (Fig. 3) associate with the peroxisome surface, especially in *atg2* and *atg7* mutants, often as a dot and sometimes as a cup or ring. They also show that peroxisome aggregates accumulate further following treatment with high-intensity light, especially in *atg* mutants (Fig. 4), and that these aggregates sometimes associate with the vacuolar membrane (Fig. 5). Finally, they show that a dye that reports ROS fluoresces more intensely in peroxisome aggregates in *atg* mutants (Fig. 6), suggesting that these peroxisome aggregates have elevated ROS levels. Although there are some interesting new data presented, the presentation and interpretation of these data could be improved.

Major concerns:

1. The authors do not consistently distinguish between what they have shown and what they hypothesize to explain their observations. For example the title states that "Pexophagy protects plants from ROS-induced damage under high-intensity light." The authors have shown that autophagy protects plants from damage caused by high-intensity light (Fig. 4a) and that peroxisomes accumulate ROS under high-intensity light when autophagy is prevented (Fig. 6a), but without a way to block pexophagy without also preventing all autophagy, it is not possible to conclude that the high-intensity light damage is caused primarily by peroxisomes that were not degraded (versus other autophagy substrates). Similar instances of are found throughout the text. For just one more example, the authors state that "ROS accumulation...is enhanced by the accumulation of inactive catalases in peroxisomes" (line 88-89), whereas the authors show that ROS accumulation is accompanied by accumulation of inactive catalases in peroxisomes. Whether accumulating ROS damages the catalase or if the ROS accumulates because catalase is damaged is not addressed.
2. The use of PEUP nomenclature burdens with reader with unnecessary additional acronyms and makes the work less accessible to readers who are not experts in plant pexophagy. The authors often identify the mutants by both aliases in the text (e.g. *peup4/atg7*), but default to only the *peup* name in the figures and abstract. It would be much preferred to use the *peup* nomenclature only when mentioning the historical identification of the mutants in the text, and then use *atg2*, *atg7*, and *atg18a* in all subsequent mentions in the text, figures, supplementary figures, tables, and videos. For example, in Supplementary Figure 15, the authors use both "*peup4*" and "*atg7*" to label different *atg7* alleles. This retention of the *peup* nomenclature is unnecessarily confusing. Using the standard *atg* nomenclature throughout the manuscript is critical for the work to be accessible to more than a handful of readers.
3. Lines 97-98: The authors indicate that *atg2*, *atg5*, *atg7*, *atg18a*, but not *atg9*, show peroxisome aggregation. These data (number of peroxisomes and frequency of aggregation) should be shown for all five mutants, because the authors show in Figure 1 that *atg7* shows smaller and less abundant aggregates than *atg2*. One might expect that *atg5* would resemble *atg7* (as both fail to lipidate ATG8) whereas *atg18a* might resemble *atg2* (gathering peroxisomes in incomplete phagophores). This demonstration of several *atg* mutants showing the same defect is essential because the authors only show results from one allele of each *atg7* and *atg2*, and thus have not demonstrated that the observed aggregation differences are due to the mutation of interest and not another unrecognized mutation in the background.
4. The data are not always internally consistent. In Sup. Fig. 14, the authors show that the *atg2*, *atg5*, *atg7*, and *atg9* mutants do not show significant peroxisome aggregation

in normal light, whereas *atg2*, *atg5*, and *atg7* show significant peroxisome aggregation in high light. Although the light intensity for the experiments in Figure 1 are not specified, it does not appear that the aggregates in *atg7* are smaller or less abundant than in *atg2* in Sup. Fig. 14. These data need to be reconciled with the data shown in Figure 1.

5. Lines 124-126: The statement "These results suggest that the *peup4/atg7* mutant accumulates high levels of ROS because of impairment in catalase activity, resulting in plant growth inhibition" is not supported by the preceding data. Catalase activity has not been measured. Moreover, the immuno-EM appears to show comparable staining of catalase in the non-dark gray areas of the peroxisomes in *wt* and *atg7* (Sup. Fig. 2). Although a higher level of catalase in the pellet in *atg7*, there is also more total catalase and no diminution of catalase in the supernatant (Sup Fig. 3). And in Sup. Figure 14c, there is no difference in the fraction of catalase in the pellet in *atg7* (or *atg2* or *atg5*) in normal versus high light.

6. The method to quantify the catalase fractionation is confusing. For example, in Figure 4f there is nearly 100% of the total catalase recovered in both the supernatant and the pellet in the *peup4/atg7* mutant grown in high light (although no indication of statistical significance is provided). The methods seem to indicate that 10 µg of protein were loaded per lane, suggesting that 10 µg of total, 10 µg of supernatant, and 10 µg of pellet protein were loaded. This would be an odd way to set up an experiment that was designed to assess the fraction of catalase that was insoluble, in which case one would load an equivalent fraction of each sample.

7. Supplemental Figure 3 is missing the loading control that was used to quantify the relative levels of peroxisomal proteins. The methods seem to indicate that mitochondrial proteins were also assayed in these experiments.

8. Supplemental Figure 5 shows a small fraction of ATG18a-GFP in the pellet in the *atg2* mutant. This experiment is missing the wild type control and the *atg7* mutant as well as a loading control. The anti-GFP blot appears to be very dirty and would benefit from an untransformed control that would validate that the correct band is being highlighted.

9. The experiments identifying ATG18a-GFP interacting proteins are problematic on several levels. The authors suggest that ATG18a-GFP is more peroxisome-associated in *atg7* than in *Wt* or *atg2* (Fig. 5), and then do a pull-down-MS experiment to find ATG18a-GFP-interacting proteins in the *atg2* mutant. The proteins recovered include various abundant enzymes from inside the peroxisome, chloroplast, and nucleus. The MS is carried out after SDS-PAGE and in-gel digestions, but it is not clear how the ATG18a-GFP and GFP control experiments were compared or which regions of the SDS-PAGE gels were analyzed. The "interacting" proteins include peroxisomal proteins, but none of these "interactions" are validated or examined in *Wt* or *atg7*. It is not clear if any of the identified proteins are direct ATG18a interactions, or how their abundance in the pull-downs compares to the abundance in the cell. This experiment is not critical to the results (it is barely mentioned in the discussion) and could be removed.

10. The PI(3)P binding reported in Supplementary Figure 11a is not convincing and could be removed.

11. The immunoEM panels in Supplementary Fig. 11b are lacking context that could be provided by a less magnified view along with the close up. It seems an overinterpretation to state that phagophore labeling is observed (line 189) based on these images.

12. The slightly darker NBT staining of *atg2* and *atg7* leaves (Sup Fig. 17a, b) following high-intensity light treatment is illustrated by photographs of smaller leaves in the mutants, which could reflect different developmental stages or cell sizes. This experiment should be repeated using leaves of the same size and include control leaves

from plants grown under normal light intensity for comparison.

13. The authors demonstrate ROS accumulation in mutant peroxisomes following high-intensity light treatment by visualizing H2-DCF fluorescence (Fig. 6, Sup. Fig. 17). These experiments are missing critical controls (staining of leaves from plants grown under normal light intensity) that are needed to conclude that this staining (and thus peroxisomal ROS) is elevated following light stress (rather than being elevated in all conditions).

14. The authors conclude that H2-DCF fluorescence intensity was higher in peroxisomes than in chloroplasts (lines 245-247) based on data in Sup. Fig. 17e. It is not clear from the figure legend or methods how this calculation was carried out and whether it takes into account the much larger cell volume occupied by chloroplasts compared to peroxisomes.

The figures can be clarified:

15. According to the instructions to authors, all of the western blots should be supported by uncropped images in a source data file. This file was not provided to the reviewer, and is especially important for figures such as Supplementary Figures 3 and 15d, in which every band is separately cropped.

16. The bar graphs would more transparently reflect the underlying data if the individual points for the biological replicates were overlaid on the plots rather than simply showing the mean and standard deviation.

17. All of the graphs in the figures and supplemental figures are missing vertical y-axis labels and instead have this information above the graph. This rearrangement is confusing to readers who expect the y-axis to be labeled. For example, the y-axis label for Fig. 1b should be "Peroxisomes/cell", 1c should be "Peroxisome aggregate frequency (%)," and 1c should be "Peroxisome aggregate size (μm^2)."

18. The bar graphs with different colored bars (e.g., Fig. 2f, 4f, S1, S3, S6, S14, S15, have text inside one set of bars to define the colors rather than the standard legend with a colored square and the definition.

19. The tri-color fluorescent images (Fig. 6a, Sup. Fig. 17c, Sup. Fig. 18a) need to show the individual channels in addition to the merge. It is not possible to interpret the overlap without the individual channels.

20. Supplementary Fig. 5, 9 – the asterisks are not useful as they seem to be highlighting the space above the band of interest. Please indicate the protein of interest with a label and a line to the band (e.g., "CAT-").

21. Supplementary Fig. 7 – please indicate units on the time stamps.

22. Supplementary Fig. 13, 14, and 15 - please indicate the allele numbers for the atg mutants assayed in the figure.

23. Supplementary Fig. 16 – the legend refers to white and gray boxes; the figure includes dark gray and light gray boxes.

24. Supplementary Table 2 – "immunoparticipation" should be "immunoprecipitation."

25. Supplementary Table 4 and 6 – Please align the column headings with the column text. Consider changing "e (SD)" to "b/a (SD)", etc.

26. Supplementary videos 11 and 12 – the scale bars are missing.

Many figure legends are missing key information necessary to understand the experiment:

27. All or most figure legends (including supplemental figures) are missing critical information such as the age of the plants analyzed, and whether they were grown in soil or on agar-based medium. Some figure legends are missing a description of the tissue or cell type imaged.

28. The early figures are missing an indication of the light intensity under which the plants were grown, which the authors later show is a critical parameter modulating peroxisome clustering in atg mutants.

29. Figure 1 legend – were peroxisomes counted per cell or were peroxisomes counted in an optical slice (how thick) of the cells? If a slice, was this near the top, bottom, or middle region of the cells (which the authors later show to be relevant)? This distinction also needs to be clarified in the methods.

30. Figure 2h legend – how long was the time-lapse image that was averaged?

31. Figure 4b-f legend – how long were the plants under high-intensity light before imaging (or running westerns)?

32. Figure 5 and 6 legend – how long were the plants under high-intensity light before imaging?

33. Supplementary video 8 legend – “white circle” should be “red circle”

34. The methods are incomplete:

35. Reference 11 describes the isolation of two *peup1/atg2* alleles; please indicate which allele was used.

36. How were the plants grown? The plants in Supplemental Figure 1 appear to be in some sort of plug inserted in the soil; this is not described in the methods or legend.

37. The methods for the lipid binding assay shown in Supplemental Figure 11a indicate the source of ATG18a-GFP (crude extract) but do not indicate the source of GFP used as a control.

The language would benefit from clarifications. For example:

38. Line 99-100 – “However, we here noticed that, in detail, the peroxisome-aggregation patterns were different between...” would be more clear as “However, the peroxisome-aggregation patterns were different between...”

39. Line 129 – “oxidative proteins” should be “oxidized proteins”

40. Line 130-131 – The authors set up the ATG8a experiments with the phrase “To examine whether ATG18a is required for selective pexophagy in plants.” However, the authors have previously shown (ref. 11) and others have confirmed (e.g., Young et al., *Autophagy* 15:941) that ATG18A is required for pexophagy. Perhaps a more accurate introduction would be “To examine ATG18a localization during pexophagy...”

41. Line 179 – consider changing “batteries” to “foci” or “accumulations”

42. Line 189 – The authors summarize Supplemental Figure 11b as “The results show that GFP-2×FYVE and ATG18a-GFP were localised on both peroxisomes and phagophores adjacent to the peroxisome.” Based on the images shown in Figure 11B, this statement would be more accurate as “The results show that GFP-2×FYVE and ATG18a-GFP were

localised both on the peroxisome periphery and adjacent to peroxisomes where the phagophore is expected to reside.”

43. Line 467 – change “fixated” to “fixed”.

44. Supplementary Fig. 19 – Please consider a different word than “undegradation.” Perhaps “Degradation failure”?

Note on editorial policy checklist:

45. The authors indicate that no unique biological materials are used in the study. This is incorrect; numerous mutant and transgenic Arabidopsis lines are used.

Reviewer #2 (Remarks to the Author):

In this paper, the authors described the contribution of the pexophagy to the selective removal of ROS-generating peroxisomes, thereby protecting plants from oxidative damage under high-intensity light. By using the cell biology and molecular biology approaches, the authors showed that the ROS-induced peroxisomes were targeted by the ATGs, eventually transporting the damage peroxisomes to the vacuole for degradation. The data and evidences are sufficient for their conclusions, which indeed extend our knowledge about the generation and formation processes of the pexophagy. However, several key issues remains to be extensively elusive. Therefore, I suggest to accept the paper after following revisions.

Major comments:

Whether the selective autophagy of the oxidative peroxisomes caused by high-intensity light is similar to that resulted from other stresses (other stresses also produce ROS)? The authors should provide sorts of evidences to distinguish them, further confirming this light-mediated pexophagy is specifically functioning in photorespiration.

If overexpressing ATGs could alleviate the oxidative peroxisomes caused by high-intensity light? These evidences should be provided to further support the notion of this paper.

If the ectopic expression of CAT could rescue the pexophagy defect in the atg mutants ?

It is still unclear who (ROS signal receptors or other unknown components?) recruit the ATG18 (and/or other ATGs?) to initiate and form the pexophagosome.

Minor comments:

The authors should carefully check the writing all over the context. A few examples are shown below:

Figures are not clear enough, especially the enlarged figure, please provide high-resolution one.

There are no statistical analyses in the figures, for example fig. 1c.

There is no annotation of materials in the supl fig 1c, and no error bar for the WT in the supl fig 1d.

199 the-- The

379 involvement-involvements

Reviewer #3 (Remarks to the Author):

In this manuscript, Oikawa et al. showed pexophagy is important for selective removal of ROS-generating peroxisomes and protects plants from oxidative damage during photosynthesis. In addition, they also showed a beautiful process of pexophagy by fluorescence labelling with GFP-2×FYVE or ATG18a-GFP and demonstrated important roles of PtdIns3P and ATG18. These findings are helpful in illustrating physiological roles of pexophagy and observation of pexophagosomes in plants, but supplied limited information about the mechanism underlying this selective autophagy process. In particular, the authors have previously demonstrated the role of pexophagy in elimination of oxidatively damaged peroxisomes in leaves and aggregation of peroxisomes in a series of mutants related to the classical ATG genes, such as ATG2, ATG18 and ATG7. Therefore, observations of peroxisome aggregates and further leaf damage in autophagy defective mutants grown under high-intensity light, which greatly induces ROS accumulation, can be expected, while, the really interesting questions in the process awaited to be answered should be how the ROS signals are recognized by pexophagy machineries or how the damaged peroxisomes are selected for degradation. Another interesting question raised by the authors, about the different pattern of peroxisome aggregation in leaf mesophyll cells between *peup1/atg2* and *peup4/atg7* mutants, is also not fully addressed, in my opinion. The authors claimed that this may reflect one ATG function in formation of pexophagosome. Does this mean the phenotypes is related to the hierarchy of ATG proteins in phagophore organization, since ATG2-ATG18 is recruited to phagophores earlier before ATG7 and determines subsequent ATG8 lipidation involving ATG7? If so, why does the aggregation phenotype in *atg7* under high-intensity light become more severe than *atg2*?

REVIEWER's COMMENTS:

Reviewer #1 (Remarks to the Author):

In this manuscript, the authors take advantage of autophagy-defective (atg) mutants and various fluorescent reporters labeling peroxisomes, ATG18a, and PIP to examine pexophagy in Arabidopsis, especially in response to high-intensity light. The authors previously reported peroxisome aggregation in atg mutants, and here show that the pattern of aggregation differs in atg2 and atg7 mutants (Fig. 1). They show that ATG18a-GFP (Fig. 2) and GFP-FYVE (Fig. 3) associate with the peroxisome surface, especially in atg2 and atg7 mutants, often as a dot and sometimes as a cup or ring. They also show that peroxisome aggregates accumulate further following treatment with high-intensity light, especially in atg mutants (Fig. 4), and that these aggregates sometimes associate with the vacuolar membrane (Fig. 5). Finally, they show that a dye that reports ROS fluoresces more intensely in peroxisome aggregates in atg mutants (Fig. 6), suggesting that these peroxisome aggregates have elevated ROS levels. Although there are some interesting new data presented, the presentation and interpretation of these data could be improved.

Major concerns:

1. The authors do not consistently distinguish between what they have shown and what they hypothesize to explain their observations. For example the title states that "Pexophagy protects plants from ROS-induced damage under high-intensity light." The authors have shown that autophagy protects plants from damage caused by high-intensity light (Fig. 4a) and that peroxisomes accumulate ROS under high-intensity light when autophagy is prevented (Fig. 6a), but without a way to block pexophagy without also preventing all autophagy, it is not possible to conclude that the high-intensity light damage is caused primarily by peroxisomes that were not degraded (versus other autophagy substrates). Similar instances of are found throughout the text. For just one more example, the authors state that "ROS accumulation...is enhanced by the accumulation of

inactive catalases in peroxisomes” (line 88-89), whereas the authors show that ROS accumulation is accompanied by accumulation of inactive catalases in peroxisomes. Whether accumulating ROS damages the catalase or if the ROS accumulates because catalase is damaged is not addressed.

2. The use of PEUP nomenclature burdens with reader with unnecessary additional acronyms and makes the work less accessible to readers who are not experts in plant peroxophagy. The authors often identify the mutants by both aliases in the text (e.g. *peup4/atg7*), but default to only the *peup* name in the figures and abstract. It would be much preferred to use the *peup* nomenclature only when mentioning the historical identification of the mutants in the text, and then use *atg2*, *atg7*, and *atg18a* in all subsequent mentions in the text, figures, supplementary figures, tables, and videos. For example, in Supplementary Figure 15, the authors use both “*peup4*” and “*atg7*” to label different *atg7* alleles. This retention of the *peup* nomenclature is unnecessarily confusing. Using the standard *atg* nomenclature throughout the manuscript is critical for the work to be accessible to more than a handful of readers.

3. Lines 97-98: The authors indicate that *atg2*, *atg5*, *atg7*, *atg18a*, but not *atg9*, show peroxisome aggregation. These data (number of peroxisomes and frequency of aggregation) should be shown for all five mutants, because the authors show in Figure 1 that *atg7* shows smaller and less abundant aggregates than *atg2*. One might expect that *atg5* would resemble *atg7* (as both fail to lipidate ATG8) whereas *atg18a* might resemble *atg2* (gathering peroxisomes in incomplete phagophores). This demonstration of several *atg* mutants showing the same defect is essential because the authors only show results from one allele of each *atg7* and *atg2*, and thus have not demonstrated that the observed aggregation differences are due to the mutation of interest and not another unrecognized mutation in the background.

4. The data are not always internally consistent. In Sup. Fig. 14, the authors show that the *atg2*, *atg5*, *atg7*, and *atg9* mutants do not show significant peroxisome

aggregation in normal light, whereas *atg2*, *atg5*, and *atg7* show significant peroxisome aggregation in high light. Although the light intensity for the experiments in Figure 1 are not specified, it does not appear that the aggregates in *atg7* are smaller or less abundant than in *atg2* in Sup. Fig. 14. These data need to be reconciled with the data shown in Figure 1.

5. Lines 124-126: The statement “These results suggest that the *peup4/atg7* mutant accumulates high levels of ROS because of impairment in catalase activity, resulting in plant growth inhibition” is not supported by the preceding data. Catalase activity has not been measured. Moreover, the immuno-EM appears to show comparable staining of catalase in the non-dark gray areas of the peroxisomes in *wt* and *atg7* (Sup. Fig. 2). Although a higher level of catalase in the pellet in *atg7*, there is also more total catalase and no diminution of catalase in the supernatant (Sup Fig. 3). And in Sup. Figure 14c, there is no difference in the fraction of catalase in the pellet in *atg7* (or *atg2* or *atg5*) in normal versus high light.

6. The method to quantify the catalase fractionation is confusing. For example, in Figure 4f there is nearly 100% of the total catalase recovered in both the supernatant and the pellet in the *peup4/atg7* mutant grown in high light (although no indication of statistical significance is provided). The methods seem to indicate that 10 µg of protein were loaded per lane, suggesting that 10 µg of total, 10 µg of supernatant, and 10 µg of pellet protein were loaded. This would be an odd way to set up an experiment that was designed to assess the fraction of catalase that was insoluble, in which case one would load an equivalent fraction of each sample.

7. Supplemental Figure 3 is missing the loading control that was used to quantify the relative levels of peroxisomal proteins. The methods seem to indicate that mitochondrial proteins were also assayed in these experiments.

8. Supplemental Figure 5 shows a small fraction of ATG18a-GFP in the pellet in the *atg2* mutant. This experiment is missing the wild type control and the *atg7*

mutant as well as a loading control. The anti-GFP blot appears to be very dirty and would benefit from an untransformed control that would validate that the correct band is being highlighted.

9. The experiments identifying ATG18a-GFP interacting proteins are problematic on several levels. The authors suggest that ATG18a-GFP is more peroxisome-associated in *atg7* than in Wt or *atg2* (Fig. 5), and then do a pull-down-MS experiment to find ATG18a-GFP-interacting proteins in the *atg2* mutant. The proteins recovered include various abundant enzymes from inside the peroxisome, chloroplast, and nucleus. The MS is carried out after SDS-PAGE and in-gel digestions, but it is not clear how the ATG18a-GFP and GFP control experiments were compared or which regions of the SDS-PAGE gels were analyzed. The “interacting” proteins include peroxisomal proteins, but none of these “interactions” are validated or examined in Wt or *atg7*. It is not clear if any of the identified proteins are direct ATG18a interactions, or how their abundance in the pull-downs compares to the abundance in the cell. This experiment is not critical to the results (it is barely mentioned in the discussion) and could be removed.

10. The PI(3)P binding reported in Supplementary Figure 11a is not convincing and could be removed.

11. The immunoEM panels in Supplementary Fig. 11b are lacking context that could be provided by a less magnified view along with the close up. It seems an overinterpretation to state that phagophore labeling is observed (line 189) based on these images.

12. The slightly darker NBT staining of *atg2* and *atg7* leaves (Sup Fig. 17a, b) following high-intensity light treatment is illustrated by photographs of smaller leaves in the mutants, which could reflect different developmental stages or cell sizes. This experiment should be repeated using leaves of the same size and include control leaves from plants grown under normal light intensity for comparison.

13. The authors demonstrate ROS accumulation in mutant peroxisomes following high-intensity light treatment by visualizing H₂-DCF fluorescence (Fig. 6, Sup. Fig. 17). These experiments are missing critical controls (staining of leaves from plants grown under normal light intensity) that are needed to conclude that this staining (and thus peroxisomal ROS) is elevated following light stress (rather than being elevated in all conditions).

14. The authors conclude that H₂-DCF fluorescence intensity was higher in peroxisomes than in chloroplasts (lines 245-247) based on data in Sup. Fig. 17e. It is not clear from the figure legend or methods how this calculation was carried out and whether it takes into account the much larger cell volume occupied by chloroplasts compared to peroxisomes.

The figures can be clarified:

15. According to the instructions to authors, all of the western blots should be supported by uncropped images in a source data file. This file was not provided to the reviewer, and is especially important for figures such as Supplementary Figures 3 and 15d, in which every band is separately cropped.

16. The bar graphs would more transparently reflect the underlying data if the individual points for the biological replicates were overlaid on the plots rather than simply showing the mean and standard deviation.

17. All of the graphs in the figures and supplemental figures are missing vertical y-axis labels and instead have this information above the graph. This rearrangement is confusing to readers who expect the y-axis to be labeled. For example, the y-axis label for Fig. 1b should be "Peroxisomes/cell", 1c should be "Peroxisome aggregate frequency (%)," and 1c should be "Peroxisome aggregate size (μm^2)."

18. The bar graphs with different colored bars (e.g., Fig. 2f, 4f, S1, S3, S6, S14,

S15, have text inside one set of bars to define the colors rather than the standard legend with a colored square and the definition.

19. The tri-color fluorescent images (Fig. 6a, Sup. Fig. 17c, Sup. Fig. 18a) need to show the individual channels in addition to the merge. It is not possible to interpret the overlap without the individual channels.

20. Supplementary Fig. 5, 9 – the asterisks are not useful as they seem to be highlighting the space above the band of interest. Please indicate the protein of interest with a label and a line to the band (e.g., “CAT-“).

21. Supplementary Fig. 7 – please indicate units on the time stamps.

22. Supplementary Fig. 13, 14, and 15 - please indicate the allele numbers for the atg mutants assayed in the figure.

23. Supplementary Fig. 16 – the legend refers to white and gray boxes; the figure includes dark gray and light gray boxes.

24. Supplementary Table 2 – “immunoparticipation” should be “immunoprecipitation.”

25. Supplementary Table 4 and 6 – Please align the column headings with the column text. Consider changing “e (SD)” to “b/a (SD)”, etc.

26. Supplementary videos 11 and 12 – the scale bars are missing.

Many figure legends are missing key information necessary to understand the experiment:

27. All or most figure legends (including supplemental figures) are missing critical information such as the age of the plants analyzed, and whether they were grown in soil or on agar-based medium. Some figure legends are missing a description

of the tissue or cell type imaged.

28. The early figures are missing an indication of the light intensity under which the plants were grown, which the authors later show is a critical parameter modulating peroxisome clustering in atg mutants.

29. Figure 1 legend – were peroxisomes counted per cell or were peroxisomes counted in an optical slice (how thick) of the cells? If a slice, was this near the top, bottom, or middle region of the cells (which the authors later show to be relevant)? This distinction also needs to be clarified in the methods.

30. Figure 2h legend – how long was the time-lapse image that was averaged?

31. Figure 4b-f legend – how long were the plants under high-intensity light before imaging (or running westerns)?

32. Figure 5 and 6 legend – how long were the plants under high-intensity light before imaging?

33. Supplementary video 8 legend – “white circle” should be “red circle”

34. The methods are incomplete:

35. Reference 11 describes the isolation of two *peup1/atg2* alleles; please indicate which allele was used.

36. How were the plants grown? The plants in Supplemental Figure 1 appear to be in some sort of plug inserted in the soil; this is not described in the methods or legend.

37. The methods for the lipid binding assay shown in Supplementary Figure 11a indicate the source of ATG18a-GFP (crude extract) but do not indicate the source of GFP used as a control.

The language would benefit from clarifications. For example:

38. Line 99-100 – “However, we here noticed that, in detail, the peroxisome-aggregation patterns were different between...” would be more clear as “However, the peroxisome-aggregation patterns were different between...”

39. Line 129 – “oxidative proteins” should be “oxidized proteins”

40. Line 130-131 – The authors set up the ATG8a experiments with the phrase “To examine whether ATG18a is required for selective pexophagy in plants.” However, the authors have previously shown (ref. 11) and others have confirmed (e.g., Young et al., *Autophagy* 15:941) that ATG18A is required for pexophagy. Perhaps a more accurate introduction would be “To examine ATG18a localization during pexophagy...”

41. Line 179 – consider changing “batteries” to “foci” or “accumulations”

42. Line 189 – The authors summarize Supplemental Figure 11b as “The results show that GFP-2×FYVE and ATG18a-GFP were localised on both peroxisomes and phagophores adjacent to the peroxisome.” Based on the images shown in Figure 11B, this statement would be more accurate as “The results show that GFP-2×FYVE and ATG18a-GFP were localised both on the peroxisome periphery and adjacent to peroxisomes where the phagophore is expected to reside.”

43. Line 467 – change “fixated” to “fixed”.

44. Supplementary Fig. 19 – Please consider a different word than “undegradation.” Perhaps “Degradation failure”?

Note on editorial policy checklist:

45. The authors indicate that no unique biological materials are used in the study. This is incorrect; numerous mutant and transgenic Arabidopsis lines are used.

Reviewer #2 (Remarks to the Author):

In this paper, the authors described the contribution of the pexophagy to the selective removal of ROS-generating peroxisomes, thereby protecting plants from oxidative damage under high-intensity light. By using the cell biology and molecular biology approaches, the authors showed that the ROS-induced peroxisomes were targeted by the ATGs, eventually transporting the damaged peroxisomes to the vacuole for degradation. The data and evidences are sufficient for their conclusions, which indeed extend our knowledge about the generation and formation processes of the pexophagy. However, several key issues remain to be extensively elusive. Therefore, I suggest to accept the paper after following revisions.

Major comments:

Whether the selective autophagy of the oxidative peroxisomes caused by high-intensity light is similar to that resulted from other stresses (other stresses also produce ROS)? The authors should provide sorts of evidences to distinguish them, further confirming this light-mediated pexophagy is specifically functioning in photorespiration.

If overexpressing ATGs could alleviate the oxidative peroxisomes caused by high-intensity light? These evidences should be provided to further support the notion of this paper.

If the ectopic expression of CAT could rescue the pexophagy defect in the atg mutants ?

It is still unclear who (ROS signal receptors or other unknown components?) recruit the ATG18 (and/or other ATGs?) to initiate and form the pexophagosome.

Minor comments:

The authors should carefully check the writing all over the context. A few examples are shown below:

Figures are not clear enough, especially the enlarged figure, please provide high-resolution one.

There are no statistical analyses in the figures, for example fig. 1c.

There is no annotation of materials in the suppl fig 1c, and no error bar for the WT in the suppl fig 1d.

199 the-- The

379 involvement-involvements

Reviewer #3 (Remarks to the Author):

In this manuscript, Oikawa et al. showed pexophagy is important for selective removal of ROS-generating peroxisomes and protects plants from oxidative damage during photosynthesis. In addition, they also showed a beautiful process of pexophagy by fluorescence labelling with GFP-2×FYVE or ATG18a-GFP and demonstrated important roles of PtdIns3P and ATG18. These findings are helpful in illustrating physiological roles of pexophagy and observation of pexophagosomes in plants, but supplied limited information about the mechanism underlying this selective autophagy process. In particular, the authors have previously demonstrated the role of pexophagy in elimination of oxidatively damaged peroxisomes in leaves and aggregation of peroxisomes in a series of mutants related to the classical ATG genes, such as ATG2, ATG18 and ATG7. Therefore, observations of peroxisome aggregates and further leaf damage in autophagy defective mutants grown under high-intensity light, which greatly induces

ROS accumulation, can be expected, while, the really interesting questions in the process awaited to be answered should be how the ROS signals are recognized by pexophagy machineries or how the damaged peroxisomes are selected for degradation. Another interesting question raised by the authors, about the different pattern of peroxisome aggregation in leaf mesophyll cells between *peup1/atg2* and *peup4/atg7* mutants, is also not fully addressed, in my opinion. The authors claimed that this may reflect one ATG function in formation of pexophagosome. Does this mean the phenotypes is related to the hierarchy of ATG proteins in phagophore organization, since ATG2-ATG18 is recruited to phagophores earlier before ATG7 and determines subsequent ATG8 lipidation involving ATG7? If so, why does the aggregation phenotype in *atg7* under high-intensity light become more severe than *atg2*?

Response to reviewers

We thank the reviewers (#1,#2, and #3) for carefully reading our manuscript and giving critical comments and pointing out misleading sentences and mistakes to improving our manuscript. We have improved the manuscript according to the reviewers' comments point by point with additional experiments and data analyzes. Please find our responses described below under each reviewers' comments.

<Reviewer #1>

Comment 1: (1) The authors do not consistently distinguish between what they have shown and what they hypothesize to explain their observations. For example the title states that “Pexophagy protects plants from ROS-induced damage under high-intensity light.” The authors have shown that autophagy protects plants from damage caused by high-intensity light (Fig. 4a) and that peroxisomes accumulate ROS under high-intensity light when autophagy is prevented (Fig. 6a), but without a way to block pexophagy without also preventing all autophagy, it is not possible to conclude that the high-intensity light damage is caused primarily by peroxisomes that were not degraded (versus other autophagy substrates).

(2) Similar instances of are found throughout the text. For just one more example, the authors state that “ROS accumulation...is enhanced by the accumulation of inactive catalases in peroxisomes” (line 88-89), whereas the authors show that ROS accumulation is accompanied by accumulation of inactive catalases in peroxisomes. Whether accumulating ROS damages the catalase or if the ROS accumulates because catalase is damaged is not addressed.

Response 1: (1) We have changed the title to “**Pexophagy suppresses ROS-induced damage in leaf cells under high-intensity light**” to not impress readers that high-intensity light damage is caused primarily by peroxisomes that were not degraded. Our

present analysis would not be enough to conclude that pexophagy is a primary factor in protecting the plant from high-intensity light-induced damage. However, our analysis clearly showed that pexophagy occurred in the leaves, based on specific targeting of ATG18a-GFP to the peroxisomes. The defect of this causes the increase of peroxisome number, peroxisome aggregation, and subsequent accumulation of ROS. These results will be enough to support that pexophagy suppresses leaf damage in the photorespiratory condition in high-intensity light.

Response 1: (2) We think that “the ROS accumulates because catalase is damaged” because previous works revealed that inactive catalase is highly accumulated in undegraded peroxisome in *atg2(p1)*, and if catalase is undamaged, ROS is removed in light condition.

We have changed the sentence on **page 5, lines 98** as follows;

[which is accompanied by the accumulation of damaged catalases in peroxisomes].

Comment 2: The use of PEUP nomenclature burdens with reader with unnecessary additional acronyms and makes the work less accessible to readers who are not experts in plant pexophagy. The authors often identify the mutants by both aliases in the text (e.g. *peup4/atg7*), but default to only the *peup* name in the figures and abstract. It would be much preferred to use the *peup* nomenclature only when mentioning the historical identification of the mutants in the text, and then use *atg2*, *atg7*, and *atg18a* in all subsequent mentions in the text, figures, supplementary figures, tables, and videos. For example, in Supplementary Figure 15, the authors use both “*peup4*” and “*atg7*” to label different *atg7* alleles. This retention of the *peup* nomenclature is unnecessarily confusing. Using the standard *atg* nomenclature throughout the manuscript is critical for the work to be accessible to more than a handful of readers.

Response 2: We have changed all the corresponding nomenclatures to standard ones following the reviewer’s comment. *peup1-1* to *atg2(p1)*, *peup2* to *atg18a(p2)*, and *peup4* to *atg7(p4)*.

Comment 3: Lines 97-98: The authors indicate that *atg2*, *atg5*, *atg7*, *atg18a*, but not *atg9*, show peroxisome aggregation. These data (number of peroxisomes and frequency of aggregation) should be shown for all five mutants, because the authors show in Figure 1 that *atg7* shows smaller and less abundant aggregates than *atg2*. One might expect that *atg5* would resemble *atg7* (as both fail to lipidate ATG8) whereas *atg18a* might resemble *atg2* (gathering peroxisomes in incomplete phagophores). This demonstration of several *atg* mutants showing the same defect is essential because the authors only show results from one allele of each *atg7* and *atg2*, and thus have not demonstrated that the observed aggregation differences are due to the mutation of interest and not another unrecognized mutation in the background.

Response 3: We have measured the number of peroxisomes and frequency of aggregation in all five mutants, including alleles of *atg2*, *atg7*, and *atg18a*. We have inserted all five-mutants data as Supplementary Figure 1. The data indicate that *atg2*, *atg5*, *atg7*, *atg18a* increased peroxisome number and aggregation compared to wild type. It suggests that the phenotype is due to the defect of autophagy genes but not another unrecognized mutation in the background. The peroxisome number and aggregation of *atg18a* and *atg5* were similar levels to *atg2* but not *atg7*, suggesting a strong effect of *atg7* deficiency in Pexophagy under normal light conditions. We have inserted the sentence on **page 6, lines 110-115**, as follow;

[However, the peroxisome-aggregation patterns were differences between the *atg* mutants at 100 $\mu\text{mol m}^{-2} \text{s}^{-1}$ (Fig. 1a–d, Supplementary Fig.1, and Supplementary Videos 1–3). The number of peroxisomes and peroxisome aggregates in *atg7* was higher than that in *atg2* and the other *atg* mutants (Fig. 1b, Supplementary Fig. 1), while the size of peroxisome aggregates (Fig. 1d) was less in *atg7* than that in *atg2*. Each allele of *atg* mutants revealed the same results (Supplementary Fig.1).]

, and on **pages 19-20, lines 411-416;**

[Under normal-intensity light, *atg7(p4)* mutant showed a higher number of peroxisome aggregates compared to other *atg* mutants (Fig. 1 and Supplementary Fig. 1). The difference may reflect the function of ATG2 and ATG7 protein in pexophagosome formation (Figs. 2, 6 and Supplementary Figs. 8, 27). No difference was observed in the degree of peroxisome aggregation under high-intensity light between *atg2* and *atg7*, suggesting that ATG2 and ATG7 play equally important roles in pexophagy.]

Comment 4: The data are not always internally consistent. In Sup. Fig. 14, the authors show that the *atg2*, *atg5*, *atg7*, and *atg9* mutants do not show significant peroxisome aggregation in normal light, whereas *atg2*, *atg5*, and *atg7* show significant peroxisome aggregation in high light. Although the light intensity for the experiments in Figure 1 are not specified, it does not appear that the aggregates in *atg7* are smaller or less abundant than in *atg2* in Sup. Fig. 14. These data need to be reconciled with the data shown in Figure 1.

Response 4: We evaluated the size and frequency of peroxisome aggregates (Supplementary Figs 14, 15 moved to **Supplementary Figs 17, 18**). The Sup. Fig. 14 c (moved to **Supplementary Figs 17c**) reveals only the size change of peroxisome aggregates between low and high-intensity light. The light intensity used for the experiments in Figure 1 and Supplementary Fig. 1 are normal-white light ($100 \mu\text{mol m}^{-2} \text{s}^{-1}$) and in Supplementary **Figs. 17, 18** (Supplementary Figs.14,15 previous) are normal- and high- intensity light from a LED equipment (blue and red light). The mutants showed different phenotypes regarding size of peroxisome aggregates in the normal-intensity white light, presumably due to the different light sources we used between the experiments.

We have done the same experiment again and concluded that peroxisome aggregates in *atg2* are similar to those in *atg7* in high-intensity light (**Supplementary Figs 17 and 18**). We have added the information of the light intensity in the each Figure legends. We have discussed the different phenotypes in the peroxisome aggregation between these mutants. Please find the sentences in **Response 3**.

Comment 5: Lines 124-126: The statement “These results suggest that the *peup4/atg7* mutant accumulates high levels of ROS because of impairment in catalase activity, resulting in plant growth inhibition” is not supported by the preceding data. Catalase activity has not been measured. Moreover, the immuno-EM appears to show comparable staining of catalase in the non-dark gray areas of the peroxisomes in wt and *atg7* (Sup. Fig. 2). Although a higher level of catalase in the pellet in *atg7*, there is also more total catalase and no diminution of catalase in the supernatant (Sup Fig. 3). And in Sup. Figure 14c, there is no difference in the fraction of catalase in the pellet in *atg7* (or *atg2* or *atg5*) in normal versus high light.

Response 5: We previously showed that peroxisome aggregates in *atg2(p1)* have inactive catalase and oxidative peroxisomes (Shibata et al, 2013). We have shown that ATG18a accumulates on the peroxisome aggregates targeted by ATG8, suggesting that the peroxisome aggregates shown in this study has the same as the previous one. We have agreed to the reviewer’s comment and changed the sentence on page 7, lines 136-138 as follows,

[These results suggest that the *atg7(p4)* mutant would accumulate high ROS levels with a high accumulation of damaged catalase in peroxisomes, resulting in plant growth inhibition.]

Comment 6: The method to quantify the catalase fractionation is confusing. For example, in Figure 4f there is nearly 100% of the total catalase recovered in both the supernatant and the pellet in the *peup4/atg7* mutant grown in high light (although no indication of statistical significance is provided). The methods seem to indicate that 10 µg of protein were loaded per lane, suggesting that 10 µg of total, 10 µg of supernatant, and 10 µg of pellet protein were loaded. This would be an odd way to set up an experiment that was designed to assess the fraction of catalase that was insoluble, in which case

one would load an equivalent fraction of each sample.

Response 6: We have changed the quantification of catalases in supernatants and pellets from the band intensity in the western blotting. In the modified quantification, we normalized the catalase levels with equivalent volume in each fraction. We have explained this with the additional sentence on **page 26, Lines 557-560** in Material and Methods. We have restored the western blotting in Figure 4. The data the quantification were provided as the Source data file.

[Signal intensities of bands in the immunoblot image were quantified using Dot Blot Analysis in Fiji. The catalase amounts in the supernatant and pellet fractions were calculated with volume-based normalisation of the extraction buffer (Source Data: Fig.4 and Supplementary Figs.18, 20).]

Comment 7: Supplemental Figure 3 is missing the loading control that was used to quantify the relative levels of peroxisomal proteins. The methods seem to indicate that mitochondrial proteins were also assayed in these experiments.

Response 7: We have attached loading control and mitochondrial proteins as the Source Data files in Supplementary **Figure 3**.

Comment 8: Supplemental Figure 5 shows a small fraction of ATG18a-GFP in the pellet in the *atg2* mutant. This experiment is missing the wild type control and the *atg7* mutant as well as a loading control. The anti-GFP blot appears to be very dirty and would benefit from an untransformed control that would validate that the correct band is being highlighted.

Response 8: We have attached the loading control to validate the band as ATG18a-GFP as the Source Data file. We have also performed immunoblotting using untransformed-wild type, wild-type control and *atg7* mutant. The anti-GFP antibody specifically

detects the ATG18a-GFP. We have attached these files in Supplementary Figure 6 with the additional sentence on **page 8, lines 157-158** as follows;

[Immunoblot showed ATG18a-GFP and catalase in the insoluble fraction of *atg2(p1)* and *atg7(p4)* (Supplementary Fig. 7a, b).]

Comment 9: The experiments identifying ATG18a-GFP interacting proteins are problematic on several levels. The authors suggest that ATG18a-GFP is more peroxisome-associated in *atg7* than in Wt or *atg2* (Fig. 5), and then do a pull-down-MS experiment to find ATG18a-GFP-interacting proteins in the *atg2* mutant. The proteins recovered include various abundant enzymes from inside the peroxisome, chloroplast, and nucleus. The MS is carried out after SDS-PAGE and in-gel digestions, but it is not clear how the ATG18a-GFP and GFP control experiments were compared or which regions of the SDS-PAGE gels were analyzed. The “interacting” proteins include peroxisomal proteins, but none of these “interactions” are validated or examined in Wt or *atg7*. It is not clear if any of the identified proteins are direct ATG18a interactions, or how their abundance in the pull-downs compares to the abundance in the cell. This experiment is not critical to the results (it is barely mentioned in the discussion) and could be removed.

Response 9: We used *atg2* because ATG18a-GFP strongly accumulates in peroxisome aggregates in *atg2* in normal-intensity light condition. We have explained how to carry out MS in Methods refer to ref 69. We have inserted additional sentences on **page 27, lines 569-571** as follows;

[The obtained proteins were electrophoresed briefly until the BPB dye band was 2 mm from the well. A 4-mm piece of gel centered on the dye band was cut out and digested with trypsin ^{76,77}.]

We have not yet examined the direct interaction between ATG18a-GFP and the candidates as the reviewer’s comment. However, we attached the list to show how

many candidates are related to the ATG18a-dependent autophagy. We will examine the function of these proteins as next works. Therefore, we would like to keep the MS analysis in this manuscript.

Comment 10: The PI(3)P binding reported in Supplementary Figure 11a is not convincing and could be removed.

Response 10: Since the analysis on PI(3)P is one of the main concerns, we would like to keep the data (moved to **Supplementary Figure 14**). The PI(3)P labeling data are convincing as the similar experiment reported in other species such as yeast (Reference 30, Krick et al, 2006; Reference 31 Tamura et al, 2013).

Comment 11: The immunoEM panels in Supplementary Fig. 11b are lacking context that could be provided by a less magnified view along with the close up. It seems an overinterpretation to state that phagophore labeling is observed (line 189) based on these images.

Response 11: Following the reviewer's comment, we inserted additional **Supplementary Fig. 14** for the panels with a large field of view to visualize the phagophore labeling to be precise.

Comment 12: The slightly darker NBT staining of atg2 and atg7 leaves (Sup Fig. 17a, b) following high-intensity light treatment is illustrated by photographs of smaller leaves in the mutants, which could reflect different developmental stages or cell sizes. This experiment should be repeated using leaves of the same size and include control leaves from plants grown under normal light intensity for comparison.

Response 12: Following reviewer's comment, we have examined NBT-stained with the leaves using the same stage under normal- and high- intensity light. We have put the photograph and statistical test in Sup Fig. 17a, b (moved to **Supplementary Figure 22**).

Comment 13: The authors demonstrate ROS accumulation in mutant peroxisomes following high-intensity light treatment by visualizing H₂-DCF fluorescence (Fig. 6, Sup. Fig. 17). These experiments are missing critical controls (staining of leaves from plants grown under normal light intensity) that are needed to conclude that this staining (and thus peroxisomal ROS) is elevated following light stress (rather than being elevated in all conditions).

Response 13: Following the reviewer's comment, we have inserted images and quantification of the H₂-DCF fluorescence examined under the low-intensity light condition as Fig 6 and Sup. Fig. 17c,d (moved to **Supplementary Figure 22**) with the additional sentence on **page 13, lines 265-267** as follows;

[H₂-DCF fluorescence was detected inside peroxisomes in the mutants with approximately two-fold higher intensity than that in wild type under high-intensity light and elevated than that under normal-intensity light (Fig. 6b and Supplementary Fig. 22c, d).]

Comment 14: The authors conclude that H₂-DCF fluorescence intensity was higher in peroxisomes than in chloroplasts (lines 245-247) based on data in Sup. Fig. 17e. It is not clear from the figure legend or methods how this calculation was carried out and whether it takes into account the much larger cell volume occupied by chloroplasts compared to peroxisomes.

Response 14: We have calculated the average intensity of H₂-DCF fluorescence inside peroxisome and chloroplast using Image J. We measured the intensity of green fluorescence from H₂-DCF in all chloroplasts and peroxisomes in the cells

surrounding with the “Polygon selections tool” in Image J. We have inserted the sentence on **page 25, lines 541-544** as follows;

[The mean intensity of fluorescence from H2-DCF inside peroxisome and chloroplast was measured using Image J. The area of peroxisome and chloroplast was determined by surrounding them with the “Polygon selections tool” in Image J.]

The figures can be clarified:

Comment 15: According to the instructions to authors, all of the western blots should be supported by uncropped images in a source data file. This file was not provided to the reviewer, and is especially important for figures such as Supplementary Figures 3 and 15d, in which every band is separately cropped.

Response 15: We have attached all of the uncropped images of the western blots in a source data file. “Supplementary Figures 3 and 15d” moved to “**Supplementary Figures 4 and 17d**” We have deleted quantification data in previous Sup. Fig 14 for avoiding repetition.

Comment 16: The bar graphs would more transparently reflect the underlying data if the individual points for the biological replicates were overlaid on the plots rather than simply showing the mean and standard deviation.

Response 16: We have arranged all the graphs to visualize individual points following the reviewer’s comments.

Comment 17: All of the graphs in the figures and supplemental figures are missing vertical y-axis labels and instead have this information above the graph. This rearrangement is confusing to readers who expect the y-axis to be

labeled. For example, the y-axis label for Fig. 1b should be “Peroxisomes/cell”, 1c should be “Peroxisome aggregate frequency (%)”, and 1c should be “Peroxisome aggregate size (μm^2).”

Response 17: Thank the reviewer for your kind suggestion. We have arranged the labels of each figure, which has the y-axis labels instead of the labels above the figures.

Comment 18: The bar graphs with different colored bars (e.g., Fig. 2f, 4f, S1, S3, S6, S14, S15, have text inside one set of bars to define the colors rather than the standard legend with a colored square and the definition.

Response 18: We have arranged the figures (e.g., Fig. 2f, 4f, S1, S3, S6, S14, S15) to be more transparent, reflecting the underlying data following the reviewer’s comment 16. Therefore, these graphs have been changed with white and black colors.

Comment 19: The tri-color fluorescent images (Fig. 6a, Sup. Fig. 17c, Sup. Fig. 18a) need to show the individual channels in addition to the merge. It is not possible to interpret the overlap without the individual channels.

Response 19: We have arranged the figures to show tri-color fluorescent images following the reviewer’s comment. We have inserted the individual channel images at Fig. 6a and Sour Date because of space limitations. “Fig. 17c and Sup. Fig. 18a” are moved to “Sup. Fig. 22c and Sup. Fig. 23a”

Comment 20: Supplementary Fig. 5, 9 – the asterisks are not useful as they seem to be highlighting the space above the band of interest. Please indicate the protein of interest with a label and a line to the band (e.g., “CAT-“).

Response 20: We have deleted the asterisks and put the label and line correspond to the band in “Supplementary Fig 5, 9” (moved to “Supplementary Fig 7, 11.”)

Comment 21: Supplementary Fig. 7 – please indicate units on the time stamps.

Response 21: We have attached “Seconds” on each column in Supplementary Figs 7,9 (moved to **Supplementary Figs 9, 12**).

Comment 22: Supplementary Fig. 13, 14, and 15 - please indicate the allele numbers for the atg mutants assayed in the figure.

Response 22: We used *atg2-1*, *atg5-1*, *atg7-2*, *atg9-3* in the Figures and have put the allele numbers in the Figures.

Comment 23: Supplementary Fig. 16 – the legend refers to white and gray boxes; the figure includes dark gray and light gray boxes.

Response 23: Thank the reviewer for pointing out our mistakes. We have arranged the figure to be black and white following the reviewer’s comment 16. We have changed the legend of Supplementary Fig.16 (moved to **Supplementary Fig. 21**).

Comment 24: Supplementary Table 2 – “immunoparticipation” should be “immunoprecipitation.”

Response 24: Thank the reviewer for pointing out our mistakes. We have corrected the error in **Supplementary Table 2**.

Comment 25: Supplementary Table 4 and 6 – Please align the column headings with the column text. Consider changing “e (SD)” to “b/a (SD)”, etc.

Response 25:

We have changed the e(SD) to “b/a (SD)”.

Comment 26: Supplementary videos 11 and 12 – the scale bars are missing.

Response 26: Thank the reviewer for pointing out the insufficient point. We have added the scale bar in these movies.

Many figure legends are missing key information necessary to understand the experiment:

Comment 27: All or most figure legends (including supplemental figures) are missing critical information such as the age of the plants analyzed, and whether they were grown in soil or on agar-based medium. Some figure legends are missing a description of the tissue or cell type imaged.

Response 27: We have added sentences about the plant age and cultural condition. We have highlighted the sentences in the legend of Figures and Supplementary Figures.

Comment 28: The early figures are missing an indication of the light intensity under which the plants were grown, which the authors later show is a critical parameter modulating peroxisome clustering in atg mutants.

Response 28: We have inserted the light intensity in the legend of Figures and Supplementary Figures.

Comment 29: Figure 1 legend – were peroxisomes counted per cell or were peroxisomes counted in an optical slice (how thick) of the cells? If a slice, was this near the top, bottom, or middle region of the cells (which the authors later

show to be relevant)? This distinction also needs to be clarified in the methods.

Response 29: We counted the peroxisome in optical slices obtained from the surface to the middle region of the cells. We have inserted the information in the Methods section and the Figure legends.

Comment 30: Figure 2h legend – how long was the time-lapse image that was averaged?

Response 30: We constructed the average image from a five-minute time-lapse image taken every 5 seconds. We have inserted the time in the legend of **Figure 2h**.

Comment 31: Figure 4b-f legend – how long were the plants under high-intensity light before imaging (or running westerns)?

Response 31: We kept the plants under high-intensity light for 16h before the imaging analysis or western blotting. We have inserted the time in the legend of **Figure 4**.

Comment 32: Figure 5 and 6 legend – how long were the plants under high-intensity light before imaging?

Response 32: We kept the plants under high-intensity light for 16 h before the imaging analysis or western blotting. We have inserted the sentence in the legend of **Figures 5 and 6**.

Comment 33: Supplementary video 8 legend – “white circle” should be “red circle”

Response 33: Thank the reviewer for pointing out our mistake. We have corrected the sentence.

The methods are incomplete:

Comment 34: Reference 11 describes the isolation of two *peup1/atg2* alleles; please indicate which allele was used.

Response 34: We used *peup1-1* as *peup1/atg2* in this study. We have inserted the *peup1-1* in the text.

Comment 35: How were the plants grown? The plants in Supplemental Figure 1 appear to be in some sort of plug inserted in the soil; this is not described in the methods or legend.

Response 35: Thank the reviewer for pointing out our insufficient sentence. We have inserted additional sentence in the methods and legend of **Supplementary Figure 1** (moved to **Supplementary Figure 2**).

Comment 36: The methods for the lipid binding assay shown in Supplementary Figure 11a indicate the source of ATG18a-GFP (crude extract) but do not indicate the source of GFP used as a control.

Response 36: We used the transgenic Arabidopsis stably expressing GFP. We have inserted the sentence on **page 27, lines 580-582** in the methods and legend (Supplementary Figure 11(moved to **Supplementary Figure 13**)).

[Crude extracts from *atg2(p1)* expressing *ATG18a-GFP* and wild type (Col-0) expressing GFP were incubated with PIP strips for 3 h at 23 °C after removing debris by centrifugation at 1,000 × *g* for 5 min.]

The language would benefit from clarifications. For example:

Comment 37: Line 99-100 – “However, we here noticed that, in detail, the peroxisome-aggregation patterns were different between...” would be more clear as “However, the peroxisome-aggregation patterns were different between...”

Response 37: We have deleted the sentence following the reviewer’s comment on **page 6, lines 110-111** as follows:

[However, the peroxisome-aggregation patterns were differences between the *atg* mutants at 100 $\mu\text{mol m}^{-2} \text{s}^{-1}$ (Fig. 1a–d, Supplementary Fig.1, and Supplementary Videos 1–3).]

Comment 38: Line 129 – “oxidative proteins” should be “oxidized proteins”

Response 38: Thank the reviewer for pointing out our mistake. We have changed the word to “oxidized” on **page 7, line 141**.

Comment 39: Line 130-131 – The authors set up the ATG18a experiments with the phrase “To examine whether ATG18a is required for selective pexophagy in plants.” However, the authors have previously shown (ref. 11) and others have confirmed (e.g., Young et al., Autophagy 15:941) that ATG18A is required for pexophagy. Perhaps a more accurate introduction would be “To examine ATG18a localization during pexophagy...”

Response 39: We agree to the reviewer’s idea and have arranged the sentence following the reviewer’s comment on **page 7, lines 142-143**.

[To examine ATG18a localisation during pexophagy in plants,]

Comment 40: Line 179 – consider changing “batteries” to “foci” or “accumulations”

Response 40: We have changed the word to “accumulations” on page 10, line 202.

Comment 41: Line 189 – The authors summarize Supplemental Figure 11b as “The results show that GFP-2×FYVE and ATG18a-GFP were localised on both peroxisomes and phagophores adjacent to the peroxisome.” Based on the images shown in Figure 11B, this statement would be more accurate as “The results show that GFP-2×FYVE and ATG18a-GFP were localised both on the peroxisome periphery and adjacent to peroxisomes where the phagophore is expected to reside.”

Response 41: We have changed the sentence following the reviewer’s suggestion on page 10, lines 194-195.

[GFP-2×FYVE and ATG18a-GFP were localised both on the peroxisome periphery and adjacent to peroxisomes where the phagophore is expected to reside.]

Comment 42: Line 467 – change “fixated” to “fixed”.

Response 42: Thank the reviewer for pointing out our mistake. We have converted the word to “fixed” on page 25, line 529.

Comment 43: Supplementary Fig. 19 – Please consider a different word than “undegradation.” Perhaps “Degradation failure”?

Response 43: We have changed the word to “undegradation” to “Degradation failure” in Supplementary Figure 19 (moved to **Supplementary Figure 27**).

Note on editorial policy checklist:

Comment 44: The authors indicate that no unique biological materials are used in the study. This is incorrect; numerous mutant and transgenic Arabidopsis lines are used.

Response 44: Thank the reviewer for pointing out our insufficient document. We used unique biological materials. We have corrected the checklist.

<Reviewer #2>

Comment 1: Whether the selective autophagy of the oxidative peroxisomes caused by high-intensity light is similar to that resulted from other stresses (other stresses also produce ROS)? The authors should provide sorts of evidences to distinguish them, further confirming this light-mediated pexophagy is specifically functioning in photorespiration.

Response 1: We have selected salt stress to distinguish pexophagy induced by high-light intensity because salt stress has been well studied to induce autophagy (Reference 36, Luo et al, 2017, Liu et al, 2009). We examined whether salt stress also induces pexophagy similar to light-mediated pexophagy. We focused on peroxisome number, aggregation, ROS accumulation, and ATG18a-GFP targeting. We concluded that the salt stress has a little effect on pexophagy, but not similar to the high-intensity light does. However, ROS was highly accumulated inside cells and peroxisomes in both wild-type and autophagy mutants, indicating that there is another mechanism of ROS accumulation. We have provided the additional **Supplementary Figures 25 and 26** with the additional sentences in Results on **pages 13-14, lines 277-282** as follow;

[We further examined the effect of salt stress^{35, 36} on pexophagy to determine whether the other types of autophagy-inducing stress mimic high-light induced pexophagy. The salt stress slightly modulated peroxisome aggregation, but to a lesser extent, despite the high accumulation of ROS in wild type and mutants (Supplementary Figs. 25, 26). The results suggest that salt stress is inefficient compared to high-light stress to induce pexophagy.]

Comment 2: If overexpressing ATGs could alleviate the oxidative peroxisomes caused by high-intensity light? These evidences should be provided to further support the notion of this paper.

Response 2: We stably overexpressed ATG18a-GFP (ATG18-OE-1 or ATG18-OE-2) in the wild-type background. We examined plant growth phenotype and ROS

accumulation in high-intensity light with the same condition of Supplementary Fig 17. Both lines revealed slight resistance to high-intensity light compared to wild type, suggesting that overexpression of ATG18 suppresses the ROS accumulation in high-intensity light. We have attached **Supplementary Figure 24** with the additional sentence in Results on **page 13, lines 275-277** as follow;

[To ascertain the contribution of autophagy in suppressing ROS in leaf cells under high-intensity light, we examined transgenic plants overexpressing ATG18a-GFP. The results revealed that ROS was suppressed in the leaf cells of the plants (Supplementary Fig. 24).]

, and in Discussion on **page 16, lines 333–336** as follow;

[We further showed that overexpression of ATG18a-GFP suppresses ROS in both cells and peroxisomes under high-intensity light (Supplementary Fig. 24), providing supporting evidence of the contribution of ATG18a in light-dependent pexophagy for preventing cell damage.]

Comment 3: If the ectopic expression of CAT could rescue the pexophagy defect in the *atg* mutants ?

Response 3: We have examined whether the ectopic expression of CAT rescues the pexophagy defect in the *atg2*. We have transiently expressed GFP-CAT and RFP-CAT in leaf mesophyll cells of *atg2* mutants visualizing peroxisome with RFP or GFP under 500 $\mu\text{mol m}^{-2}\text{s}^{-1}$. We counted the number of peroxisomes and peroxisome aggregates in leaf mesophyll cells of *atg2* and found that ectopic expression of CAT slightly rescues the pexophagy defect in *atg2* in high-intensity light. We have inserted **Supplementary Figure 19** with an additional sentence in Results on **page 11, lines 223–226** as follow;

[To examine the relationship between peroxisome aggregation and abnormal catalase accumulation in autophagy mutants under high-intensity light, we

ectopically overexpressed CAT2 in *atg2* mutant. The results showed that CAT2 suppressed peroxisome aggregation in *atg2* mutant (Supplementary Fig. 19).]

, and in Discussion on page 15, lines 314 –317 as follows;

[The ectopic expression of catalase in leaf cells of *atg2* suppressed the peroxisome number and aggregation in the cells (Supplementary Fig. 19), suggesting that active catalase would reduce peroxisomal ROS and subsequent peroxisome aggregation in *atg2*.]

Comment 4: It is still unclear who (ROS signal receptors or other unknown components?) recruit the ATG18 (and/or other ATGs?) to initiate and form the pexophagosome.

Response 4: We are also interested in the question about the receptor for ROS signal inducing pexophagy with recruiting ATG18. We showed that PtdIns3P was formed around degraded peroxisome or pexophagosome, where ATG18a localized. We have examined the effect of a PI3 kinase inhibitor wortmannin on the targeting of ATG18a-GFP and GFP-2xFYVE on the peroxisomes. We found that both ATG18a-GFP and GFP-2xFYVE remarkably lost the localization on degrade-peroxisomes, suggesting that PI3-kinase is involved in step of pexophagy. It is of great interest for the receptor to initiate pexophagosome formation, but we would like to report the components as a separate publication because finding the pexophagy receptor is challenging. We added **Supplementary Figure 16** with additional sentences about the wortmannin experiment in Results on page 10, lines 205 – 208 as follows;

[This was supported by the evidence that wortmannin^{16, 18, 21, 32}, a phosphoinositide 3-kinase inhibitor, disturbed the subcellular localisation of ATG18a-GFP and GFP-2xFYVE on peroxisome aggregates in *atg2(p1)* or *atg18a(p2)* (Supplementary Fig. 16).]

, and in Discussion on page 18, lines 373– 376 as follow;

[Disturbance of the subcellular localisation of the ATG18a-GFP and GFP-2xFYVE on degraded peroxisomes in wortmannin-treated *atg2(p1)* or *atg18a(p2)* (Supplementary Fig. 16) implies that phosphoinositide 3-kinase activity is required to form the platform for gathering ATG18a on degraded peroxisomes.]

Minor comments:

The authors should carefully check the writing all over the context. A few examples are shown below:

Response: We acknowledge the reviewer's kind advice to improve our manuscript. We corrected our manuscript by checking with English experts.

Comment 5: Figures are not clear enough, especially the enlarged figure, please provide high-resolution one.

Response 5: We have replaced the enlarged figures with high-resolution images in **Figs.1-6**.

Comment 6: There are no statistical analyses in the figures, for example fig. 1c. There is no annotation of materials in the suppl fig 1c, and no error bar for the WT in the suppl fig 1d.

Response 6: We have performed statistical analyses on the Figures. We attached n.s in **Fig.1c** because there was no significant difference between the mutants. We have also attached the material annotations and error bar in **Supplementary Fig 1d**.

Comment 7: 199 the-- The

Comment 8: 379 involvement- involvements (p19, line 400)

Response to 7 and 8: Thank you for pointing out our mistakes. We have corrected them.

<Reviewer #3>

Comments 1: In this manuscript, Oikawa et al. showed pexophagy is important for selective removal of ROS-generating peroxisomes and protects plants from oxidative damage during photosynthesis. In addition, they also showed a beautiful process of pexophagy by fluorescence labelling with GFP-2×FYVE or ATG18a-GFP and demonstrated important roles of PtdIns3P and ATG18. These findings are helpful in illustrating physiological roles of pexophagy and observation of pexophagosomes in plants, but supplied limited information about the mechanism underlying this selective autophagy process. In particular, the authors have previously demonstrated the role of pexophagy in elimination of oxidatively damaged peroxisomes in leaves and aggregation of peroxisomes in a series of mutants related to the classical ATG genes, such as ATG2, ATG18 and ATG7. Therefore, observations of peroxisome aggregates and further leaf damage in autophagy defective mutants grown under high-intensity light, which greatly induces ROS accumulation, can be expected, while, the really interesting questions in the process awaited to be answered should be how the ROS signals are recognized by pexophagy machinerie or how the damaged peroxisomes are selected for degradation.

Response 1: We thank the reviewer for carefully reading our manuscript with critical comments. The degraded peroxisomes accumulate inactive catalase and ROS. These peroxisomes are labeled with ATG18a-GFP and GFP-2xFYVE adjacent to autophagosome-like structure for the degradation by autophagy system. As additional information, we showed that wortmannin disturbed the localization of ATG18a-GFP and GFP-2xFYVE on the degraded peroxisomes (**Supplementary Figure 16**), suggesting that PI3-kinase plays a role in the localization of ATG18 and PtdIns3P. We also found that CFP-ATG8 accumulates with ATG18a-GFP on the peroxisome aggregations in *atg2*, suggesting that PtdIns3P formation around degraded peroxisomes is an early step to accumulate ATG8 for autophagic degradation of peroxisomes. We are also interested in the question about the receptor for ROS signal inducing pexophagy with recruiting ATG18. Although we do not have any conclusive evidence about the component that senses ROS signal for inducing pexophagy, we are

close to answering the question. We performed MS analysis for finding candidate proteins binding to ATG18a-GFP on peroxisomes in *atg2* (**Supplementary Fig. 11 and Supplementary Table 2**). We are expecting that there are sensors or receptors for pexophagy in the list. However, we would like to report the components as a separate publication because finding the pexophagy receptor is challenging.

We have inserted additional sentences about the experiment of the wortmannin and ATG8 as follows;

about the wortmannin in Results on **page 10, lines 205 – 208**;

[This was supported by the evidence that wortmannin^{16,18,21,32}, a phosphoinositide 3-kinase inhibitor, disturbed the subcellular localisation of ATG18a-GFP and GFP-2xFYVE on peroxisome aggregates in *atg2(p1)* or *atg18a(p2)* (Supplementary Fig. 16).]

, and in Discussion on **page 18, lines 373– 376**;

[Disturbance of the subcellular localisation of the ATG18a-GFP and GFP-2xFYVE on degraded peroxisomes in wortmannin-treated *atg2(p1)* or *atg18a(p2)* (Supplementary Fig. 16) implies that phosphoinositide 3-kinase activity is required to form the platform for gathering ATG18a on degraded peroxisomes.]

We have inserted the sentences about ATG8e in Results on **page 8, lines 152-156**;

[We have previously shown that ATG8 accumulates near the peroxisome aggregates in *atg2(p1)*^{11,14}. To examine whether ATG18 and ATG8 targets to the same peroxisome aggregate in *atg2(p1)*, we transiently expressed CFP-ATG8e in *atg2(p1)* expressing ATG18a-GFP. The result showed that CFP-ATG8e and ATG18a-GFP are colocalised to the same peroxisome aggregate (Supplementary Fig. 6), revealing that ATG18a recognises oxidized peroxisomes to be degraded.]

, and in Discussion on **page 18, lines 386-389**;

[We have previously revealed that ATG8a localises to degraded peroxisomes in *atg2(p1)* and *atg5* as dot structures^{11, 14}. In this study, we found that ATG8e colocalises with ATG18a on peroxisome aggregates in *atg2(p1)* (Supplementary Fig. 6), suggesting that ATG8 acts in concert with ATG18a.]

and, page 20, lines 431-435 as follows;

[We showed that phosphoinositide 3-kinase is involved in pexophagosome formation (Supplementary Fig. 16). The future direction of this study is to find the sensor protein(s) sensing ROS or oxidised lipids on the peroxisome membrane for activating the phosphoinositide 3-kinase to induce pexophagy and clarify the involvement of *PEXs* in plant pexophagy.]

Comments 2: Another interesting question raised by the authors, about the different pattern of peroxisome aggregation in leaf mesophyll cells between *peup1/atg2* and *peup4/atg7* mutants, is also not fully addressed, in my opinion. The authors claimed that this may reflect one ATG function in formation of pexophagosome. Does this mean the phenotypes is related to the hierarchy of ATG proteins in phagophore organization, since ATG2-ATG18 is recruited to phagophores earlier before ATG7 and determines subsequent ATG8 lipidation involving ATG7? If so, why does the aggregation phenotype in *atg7* under high-intensity light become more severe than *atg2*?

Response 2: We have reanalyzed the phenotype of *atgs* mutants under the different light intensity and concluded that the different phenotype of the peroxisome aggregation was observed only in normal-intensity light (Supplementary Fig. 1) but not in high-intensity light (Supplementary Figs. 17,18). As the reviewer's comments, the difference in phenotype of peroxisome aggregate (size, frequency) would be related to ATG function such as pexophagosome formation. We have shown that the number of the ring structure of ATG18a-GFP in *atg7* is higher than that in *atg2* in normal-intensity light (Fig. 2f and Supplementary Fig. 8). However, in high-intensity light, undegraded peroxisome increase to form large aggregation because degradation of

peroxisomes in vacuole could not proceed both in *atg2* and *atg7* mutants. We have inserted an additional sentence about the different phenotype of the peroxisome aggregation between in *atg2* and *atg7* on **pages 19-20, lines 411-416** as follow;

[Under normal-intensity light, *atg7(p4)* mutant showed a higher number of peroxisome aggregates compared to other *atg* mutants (Fig. 1 and Supplementary Fig. 1). The difference may reflect the function of ATG2 and ATG7 protein in pexophagosome formation (Figs. 2, 6 and Supplementary Figs. 8, 27). No difference was observed in the degree of peroxisome aggregation under high-intensity light between *atg2* and *atg7*, suggesting that ATG2 and ATG7 play equally important roles in pexophagy.]

Reviewer #1 (Remarks to the Author):

I have read the revised manuscript and find that the authors have satisfactorily addressed my previous concerns. Additional experiments have been added that strengthen the conclusions, the figures are improved, and the data are more clearly described.

I have only minor suggestions for improvement:

Line 82 – “leading to the oxidisation of peroxisomes. We have shown that oxidative peroxisomes are not degraded in autophagy-deficient mutants” would be more clear as “leading to the oxidation of peroxisomes. We have shown that oxidation-damaged peroxisomes are not degraded in autophagy-deficient mutants.” (All peroxisomes are presumably oxidative.)

Line 110 – “differences” should be “different”

Line 225 – “we ectopically overexpressed CAT2 in atg2 mutant. The results showed that CAT2 suppressed peroxisome aggregation in atg2 mutant” should be “we ectopically overexpressed GFP-CAT2 or RFP-CAT2 in an atg2 mutant. We found that these CAT2 fusions suppressed peroxisome aggregation in atg2 mutant.”

Line 260 – “accumulat” should be “accumulate”

Supplementary Figure 1 title – “atgs mutants” should be “atg mutants”

Supplementary Figure 4 – the legend does not match the figure. Panel b in the legend refers to immunoblotting of a fractionation experiment; panel b of the figure is quantification of panel a.

Reviewer #2 (Remarks to the Author):

All the issues raised by reviewers have been addressed by the authors, and the quality of the manuscript has significantly been improved. Therefore, I suggested to accept this paper.

Reviewer #3 (Remarks to the Author):

In this work, the authors carried out a lot of imaging studies on peroxisome status in wild-type *Arabidopsis* as well as atg mutants under high-intensity light, aiming to show that pexophagy was induced during the strong light irradiation to eliminate damaged peroxisomes, and failure of the pexophagy in atg mutants caused more severe aggregation of peroxisomes and leaf damage. However, since the phenotypes involving peroxisome aggregates, catalase inactivation and ROS accumulation in atg mutants grown under normal conditions had been reported by the same research group, a series of data (such as Figure 1,4, and Supplementary Figure 1-4, 17,18,22) showing the those kinds of phenotypes in autophagy defective mutants under the high-intensity light, which is actually just an enhanced oxidative stress to plants in the nature, can be expected and provided little new information. The only new and interesting message is about the co-labeling of ATG18a-GFP or GFP-2×FYVE with some peroxisomes, which might be an indicator for pexophagosomes and pexophagy, if the connections between the appearance of ATG18a/ FYVE on peroxisomes and the destinies of the marked peroxisomes are convincingly proved. Unfortunately, however, the current data are obviously not enough.

1. The authors mainly focused on the observation of increased associations of ATG18a-GFP or GFP-2×FYVE with aggregated peroxisomes in autophagic-defective mutants, especially under normal light conditions in most cases, but did not pay real attention to the molecular links between the pexophagy process induced by the high-intensity light and the dynamic changes of co-localizations of aggregated peroxisomes with ATG18a-GFP or GFP-2×FYVE, the morphology changes of the co-labeled structures, and the incorporation process by vacuole during pexophagy. In addition, although various kinds of structures, labeled by ATG18a-GFP or GFP-2×FYVE (like dot, cup and

ring-like structures), resembling pexophagic structures at different steps, were observed, their identities still need to be further proved. Actually, the current imaging data only tell the possible enrichment of PI3P and ATG18a-GFP on aggregated peroxisomes, but cannot convince readers to believe that those ATG18a-GFP- or GFP-2×FYVE-positive structures associated with peroxisomes are surely pexophagic structures. Co-localizations of ATG18a-GFP or GFP-2×FYVE with ATG8-labeled puncta should be monitored during a complete pexophagy process in the stable transgenic lines with both fluorescent markers to investigate those questions that include but not limited to the recruitment order of these proteins to the damaged peroxisome, the stages that they participated in, the duration time of co-localizations between them on peroxisomes and their destinies when pexophagosomes are formed or degraded, but not just observations of co-localizations between ATG18a-GFP with ATG8 in *atg2* mutant, or just similarity analysis of their localization patterns on aggregated peroxisomes in the current manuscript. Furthermore, PAS-like structures, indicated by the authors, in the TEM images of supplementary figure 13-14, need to be further confirmed by immunogold labeling of ATG8. In addition, evaluation of vacuolar peroxisome degradation in wild-type cells, before and after high light irradiation, is quite necessary to address the occurrence of pexophagy in plants under high-intensity light. This can be achieved by monitoring the appearance of peroxisomic structures in vacuoles or by biochemical analysis of peroxisomic protein degradation, with the help of inhibitors of vacuolar proteases or V-ATPase.

2. The light conditions used in this study for high-intensity treatments were not consistent. In most cases, 1000 $\mu\text{mol m}^{-2} \text{s}^{-1}$ was applied, but in some cases, 500 $\mu\text{mol m}^{-2} \text{s}^{-1}$ and 200 $\mu\text{mol m}^{-2} \text{s}^{-1}$ were also used. The inconsistency was also true for the *atg* mutants. In the beginning, as the author stated, the number of peroxisomes and peroxisome aggregates in *atg7* was higher and leaf damages of *atg7* under high-intensity light were also more severe than other *atg* mutants, they mainly chose *atg7(p4)* for subsequent study. However, in some of the following studies, a wealth of data, as presented in Supplementary Figure 6, 10, 11, 13, 14, 16 and 19, were obtained only from *atg2* mutant.

3. Since most of the conclusions are obtained from confocal imaging studies, the quality of confocal data is of great importance. The authors should distinguish carefully the real punctate associations of ATG18a-GFP or GFP-2×FYVE with organelles from just artifacts caused by occasional space-overlap, which is especially true for the co-localization analysis of those two proteins with chloroplasts shown in Supplementary Figure 23. All of the images to show the co-localizations of RFP-PTS1 labeled peroxisomes with ATG18a-GFP or GFP-2×FYVE or tri-labeling should also include the splitted channels for readers to observe the precise morphology of the structures. Structures in some of the images cannot be clearly visualized in the less-magnified view, e.g. Supplementary Figure 8b and 12b. Some images would be better presented by showing a large field of view along with the magnified view of some specified structures for readers to understand the frequency, subcellular locations and detailed morphologies, such as supplementary Figure 6 and especially observations of peroxisomes surrounded by vacuolar membranes in Figure 5 and Supplementary Figure 21a. In addition, the current descriptions about the relative positions of peroxisome aggregates around vacuolar membranes are confusing. Tonoplast and vacuole bubbles were not strictly distinguished and collectively referred to as vacuolar membranes. In addition, whether some peroxisome aggregates appeared within vacuoles, whether those vacuolar-incorporated peroxisome aggregates are enveloped with tonoplast-derived membranes, and the quantification of the numbers of corresponding types of peroxisome aggregates defined by the above features should be clarified, which will help the readers to better understand the occurrence of pexophagy via macroautophagy or microautophagy.

4. The authors performed a lot of quantifications of intensities related to WB and H2-DCF staining. For quantification of WB images in Supplementary Figure 18C and 20f, were the two films exposed at the same time? If not, how can the authors convince readers to believe that the intensity differences were results of signals or just exposure time? This reviewer is also curious that how the H2-DCF staining signals from different leaves of different plants can be scientifically quantified without an internal control to avoid interference of leaf thickness. In addition, the intensities quantified in Figure 6a indicate some signals were possibly out of the linear dynamic range of the detector and overexposed

5. In addition to the localizations of ATG18a-GFP or GFP-2×FYVE on some peroxisomes, where do

their other signals mainly appear at the subcellular level. Please add images to show the localization patterns of ATG18a-GFP and GFP-2×FYVE in wild-type cells under normal conditions. In supplementary figure 15d, some GFP signals of ATG18a-GFP seem to appear on chloroplasts. Is this case normal? Also, the last three panels of the supplementary figure 15d were shown in the bright field mode, while the first one is not. Those data need to be displayed in a consistent manner.

6. The authors demonstrated that ectopic expression of catalase in *atg2* mutant decreased not only the number of peroxisome aggregates but also the total number of peroxisomes. This result is quite surprising. How can the aggregated peroxisomes can be further degraded in a autophagy-defective mutant?

7. As illustrated by the authors, leaf damage under high-intensity light is closely related to catalase inactivation, peroxisome aggregation and excess ROS accumulation. However, although the authors claimed that overexpression of ATG18 significantly suppressing accumulation of ROS within mesophyll cells, the ATG18 OE lines did not show visible alleviated leaf damage under high-intensity light. Besides, the leaf damage in wild-type plants subjected to high light treatment, as shown in Supplementary Figure 17a, seems to be second only to *atg7* mutant, because bleach phenotypes can be both observed in those two plants, but not in other *atg* mutants.

8. Data involving mitochondrial degradation and chloroplast damage are very preliminary and has little to do with the main scientific question need to be solved in this manuscript, which can be removed from the maintext.

9. Supplementary figure 15c and d were not referenced in the main text.

10. Internal control for supplementary figure 4a deposited in source data should be assembled at the bottom of the figure. Similar analysis in supplementary figure 20d lacks internal control.

11. Figure legend of supplementary figure 4b is not correct.

12. The TEM image in Supplementary figure 20b is meaningless without being accompanied with the proper control.

13. The pseudocolors for peroxisomes and chloroplast should be different.

14. Splitted images from individual channels for the up panel of Supplementary Figure 22c are missing in the source data.

15. Typo errors in source data:

Supplementary Figure 20f, the labels for antibody of the bottom panel should be anti-SHMT

Supplementary Figure 21b, High light (100 $\mu\text{molm}^{-2}\text{s}^{-1}$) should be High light (1000 $\mu\text{molm}^{-2}\text{s}^{-1}$)

16. The normalization methods for intensity analysis in source data were not consistent:

Raw data for Fig. 4f, Sfig.22b and Sfig.24d were normalized to the wild-type plants, whose mean was set to 1.

Raw data for SFig.2e and Sfig.24c were normalized to the wild-type plants, whose mean was not set to 1.

Raw data for SFig.4b and SFig.20e were normalized to the wild-type plants, but the value of individuals was set to 1.

Dear Reviewers (#1, #2, and #3),

We thank all reviewers for reviewing our manuscript and providing critical and constructive comments. These comments guided our manuscript in becoming more accessible to the readers. We have responded to the reviewers' comments point by point. Our response to each comment is provided under reviewers' comments. We have carried out additional experiments requested by the reviewers. The new results are included in Figures 6, Supplementary Figures 7, 12, 17, 19, 20b, 24, 27-29, 32b, and Supplementary Videos 9, 13-15, 18-22 with splits images in source data file. According to these findings, we revised the final model (Figure 7c) and sentences in Results and Discussion in the revised manuscript. The additions and corrections are indicated in red text. Additionally, the manuscript has been comprehensively reviewed and edited by a professional native English-speaking editor to meet the language standards required by leading publications.

REVIEWER COMMENTS

Reviewer #1 (Remarks to the Author):

I have read the revised manuscript and find that the authors have satisfactorily addressed my previous concerns. Additional experiments have been added that strengthen the conclusions, the figures are improved, and the data are more clearly described.

I have only minor suggestions for improvement:

Response to reviewer #1

Comment 1. Line 82 – “leading to the oxidisation of peroxisomes. We have shown that oxidative peroxisomes are not degraded in autophagy-deficient mutants” would be clearer as “leading to the oxidation of peroxisomes. We

have shown that oxidation-damaged peroxisomes are not degraded in autophagy-deficient mutants.” (All peroxisomes are presumably oxidative.)

Response 1 We changed the sentence following the reviewer’s comment on Pages 3, lines 65–68 as follows,

[ROS accumulation in peroxisomes inhibits catalase (CAT) activity that detoxifies hydrogen peroxide, leading to the oxidation of peroxisomes^{5,9,10}. We have previously shown that oxidatively damaged peroxisomes are accumulated in autophagy-deficient mutants^{11,14}.]

Comment 2. Line 110 – “differences” should be “different”

Response 2. We corrected the mistake on Page 4, line 90.

Comment 3. Line 225 – “we ectopically overexpressed CAT2 in atg2 mutant. The results showed that CAT2 suppressed peroxisome aggregation in atg2 mutant” should be “we ectopically overexpressed GFP-CAT2 or RFP-CAT2 in an atg2 mutant. We found that these CAT2 fusions suppressed peroxisome aggregation in atg2 mutant.”

Response 3. We changed the sentence on Page 9, lines 197–199 as follows:

[We overexpressed GFP-CAT2 or RFP-CAT2 to recover catalase activity and discovered that CAT2 fusion overexpression suppressed the increase in

peroxisome numbers and their aggregation in the *atg2* mutant (Supplementary Fig. 23).]

Comment 4. Line 260 – “accumulat” should be “accumulate”

Response 4. We corrected the typo on Page 12, line 261.

Comment 5. Supplementary Figure 1 title – “atgs mutants” should be “atg mutants”

Response 5. We corrected it to “atg mutants” in Supplementary Figure 1 title.

Comment 6. Supplementary Figure 4 – the legend does not match the figure.
Panel b in the legend refers to immunoblotting of a fractionation experiment;
panel b of the figure is quantification of panel a.

Response 6. Thank you for pointing out these mistakes. We have corrected the sentence in Supplementary Figure 4 b as follows:

[b, Signal intensity of immunoblotting analysis in (a).]

Reviewer #3 (Remarks to the Author):

In this work, the authors carried out a lot of imaging studies on peroxisome status in wild-type Arabidopsis as well as atg mutants under high-intensity light, aiming to show that pexophagy was induced during the strong light irradiation to eliminate damaged peroxisomes, and failure of the pexophagy in atg mutants caused more severe aggregation of peroxisomes and leaf damage. However, since the phenotypes involving peroxisome aggregates, catalase inactivation and ROS accumulation in atg mutants grown under normal conditions had been reported by the same research group, a series of data (such as Figure 1,4, and Supplementary Figure 1-4, 17,18,22) showing the those kinds of phenotypes in autophagy defective mutants under the high-intensity light, which is actually just an enhanced oxidative stress to plants in the nature, can be expected and provided little new information. The only new and interesting message is about the co-labeling of ATG18a-GFP or GFP-2×FYVE with some peroxisomes, which might be an indicator for pexophagosomes and pexophagy, if the connections between the appearance of ATG18a/ FYVE on peroxisomes and the destinies of the marked peroxisomes are convincingly proved. Unfortunately, however, the current data are obviously not enough.

Response to reviewer #3

Comment 1. The authors mainly focused on the observation of increased associations of ATG18a-GFP or GFP-2×FYVE with aggregated peroxisomes in autophagic-defective mutants, especially under normal light conditions in most cases, but did not pay real attention to the molecular links between the pexophagy process induced by the high-intensity light and the dynamic changes of co-localizations of aggregated peroxisomes with ATG18a-GFP or GFP-2×FYVE, the morphology changes of the co-labeled structures, and the incorporation process by vacuole during pexophagy. In addition, although various kinds of structures, labeled by ATG18a-GFP or GFP-2×FYVE (like

dot, cup and ring-like structures), resembling pexophagic structures at different steps, were observed, their identities still need to be further proved. Actually, the current imaging data only tell the possible enrichment of PI3P and ATG18a-GFP on aggregated peroxisomes, but cannot convince readers to believe that those ATG18a-GFP- or GFP-2×FYVE-positive structures associated with peroxisomes are surely pexophagic structures.

(1) Co-localizations of ATG18a-GFP or GFP-2×FYVE with ATG8-labeled puncta should be monitored during a complete pexophagy process in the stable transgenic lines with both fluorescent markers to investigate those questions that include but not limited to the recruitment order of these proteins to the damaged peroxisome, the stages that they participated in, the duration time of co-localizations between them on peroxisomes and their destinies when pexophagosomes are formed or degraded, but not just observations of co-localizations between ATG18a-GFP with ATG8 in *atg2* mutant, or just similarity analysis of their localization patterns on aggregated peroxisomes in the current manuscript.

(2) Furthermore, PAS-like structures, indicated by the authors, in the TEM images of supplementary figure 13-14, need to be further confirmed by immunogold labeling of ATG8.

(3) In addition, evaluation of vacuolar peroxisome degradation in wild-type cells, before and after high light irradiation, is quite necessary to address the occurrence of pexophagy in plants under high-intensity light. This can be achieved by monitoring the appearance of peroxisomic structures in vacuoles or by biochemical analysis of peroxisomic protein degradation, with the help of inhibitors of vacuolar proteases or V-ATPase.

Response 1 (1) We thank the reviewer for the critical comments and suggestions. Although we understand the reviewer's point, we could not generate stable transgenic plants to track both ATG18a and ATG8 simultaneously due to the short

revising time and the technical limitation. We observed that the ATG18a-GFP is rarely present in wild-type cells (**Figure 2b**), which implies that it would be challenging to track them even in a stable transformant. We identified ATG8 and ATG18 co-localisation on peroxisome aggregates in the *atg2* mutant (**Supplementary Figure 6**), in which the fast-midway stage of autophagy can be visible because of the mutation. We newly examined and added ATG8 and ATG18a-GFP co-localisation on peroxisome aggregation in *atg7* mutants (**Supplementary Fig. 7**). We further examined peroxisomes targeting ATG18a-GFP using the FRAP analysis (**Supplementary Figs. 11, 12**) and the effect of wortmannin on peroxisomes targeting ATG18a-GFP (**Supplementary Fig. 19**). Our results are consistent with previous results (Shibata et al, 2013 and Yoshimoto et al, 2014), with ATG18 and ATG8 co-localisation taking place on the pexophagosomes, which has been reported by other research groups (Le et al, 2014⁵³, Zhuang et al, 2017⁵⁴). Therefore, our results strongly support the participation of both ATG18 and ATG8 in pexophagy. We could identify little ATG18a-GFP targeted peroxisomes in the wild type cells. Therefore, we will track the ATG18a-GFP and ATG8a by making several transgenic lines with an inducible promoter and autophagy inhibitors to monitor pexophagy efficiently, which will be a different project, not included in this study.

- (2) To address the reviewer comment, we examined ATG8 localization on pexophagy structure in *atg2* using an antibody against ATG8 (**Supplementary Fig. 17**). The results clearly showed the ATG8 within the pexophagy structure similar to the TEM analyses of ATG18a and GFP-2xFYVE. We have provided an additional Figure (**Supplementary Fig. 17**) with the additional sentence as follows:

[These localisations were similar to that of ATG8 analysed using an anti-ATG8 antibody (Supplementary Figs. 17).] (Page 8, lines 166–167)

[Detailed analysis by electron microscopy revealed that PtdIns3P and ATG18a were localised on both peroxisomes and phagophores adjacent to peroxisomes

in *atg2(p1)* (Supplementary Figs. 15b, 16), similarly to ATG8 (Supplementary Fig. 17).] (Page 16, lines 357–359)

[PtdIns3P, ATG18a, and ATG8 were localised to the same area (Figs. 2, 3 and Supplementary Figs. 15b, 16, and 17).] (Page 16, lines 364–365)

(3) To address the reviewer's request, we examined the effect of concanamycin A (ConA), a V-ATPase inhibitor, on the low- and high-intensity light-induced peroxisome degradation in the vacuole. The peroxisomes were accumulated in the vacuole in the ConA-treated leaves exposed to high-intensity light. We added the **supplementary Figure 24 and Supplementary Videos 12-15** with an additional sentence on Page 9, lines 199–203 as follows:

[To examine vacuolar peroxisome degradation by pexophagy under high-intensity light conditions, we inhibited vacuolar H⁺-ATPase using concanamycin A (ConA) to stop vacuolar hydrolytic activity in wild-type leaf mesophyll cells (Supplementary Fig. 24). The ConA-treated cells increased the accumulation of undegraded-peroxisomes in the vacuole under high-intensity light conditions (Supplementary Fig. 24a-c and Supplementary Videos 12-15). These results indicated that pexophagy was facilitated in high-intensity light.]

and, on Page 15, lines 325–326 as follows:

[The peroxisome number was increased in the vacuole of ConA-treated wild-type cells in the high-intensity light compared to the low-intensity light (Supplementary Fig. 24a-c).]

Comment 2. (1) The light conditions used in this study for high-intensity treatments were not consistent. In most cases, 1000 $\mu\text{mol m}^{-2} \text{s}^{-1}$ was

applied, but in some cases, $500 \mu\text{mol m}^{-2} \text{s}^{-1}$ and $200 \mu\text{mol m}^{-2} \text{s}^{-1}$ were also used. (2) The inconsistency was also true for the *atg* mutants. In the beginning, as the author stated, the number of peroxisomes and peroxisome aggregates in *atg7* was higher and leaf damages of *atg7* under high-intensity light were also more severe than other *atg* mutants, they mainly chose *atg7(p4)* for subsequent study. However, in some of the following studies, a wealth of data, as presented in Supplementary Figure 6, 10, 11, 13, 14, 16 and 19, were obtained only from *atg2* mutant.

Response 2. (1) We used different intensity lights to observe the effects of light intensity on plants. We first examine the white light condition between 50 to $200 \mu\text{mol m}^{-2} \text{s}^{-1}$. We discovered peroxisome aggregation and growth reduction in *atg* mutants following exposure to 100 to $200 \mu\text{mol m}^{-2} \text{s}^{-1}$, which we recognized as normal-intensity light because we use such light condition for wild-type plant growth. We categorised 500 and $1000 \mu\text{mol m}^{-2} \text{s}^{-1}$ light as the high-intensity light using an LED equipment (blue and red), which is effective for photosynthesis, because it induces photo stress, e.g., higher peroxisome aggregation and ROS accumulation. Therefore, the light conditions can be divided into two: low-intensity and high-intensity light conditions. Small photon flux density changes within the same light condition were mainly due to instrumental availability during the project. Still, we observed consistent results within the relevance of the same light condition. For transient expression assay using the cutting leaf, protoplast and vacuole isolation experiment, we used $500 \mu\text{mol m}^{-2} \text{s}^{-1}$ as high-intensity light because a $1000 \mu\text{mol m}^{-2} \text{s}^{-1}$ light was too stressful and reduced the protoplast yield. We described the reasons for using another light intensity on **Page 3, at lines 70-71 in Supplementary Information** as follows:

[We used $500 \mu\text{mol m}^{-2} \text{s}^{-1}$ for preventing excess damage to the cutting leaves after particle bombardment.]

(2) We did the same experiment (Supplementary Figure 6) with an *atg7* mutant, showing co-localisation of ATG8 and ATG18a-GFP on peroxisome aggregation (**Supplementary Figure. 7**).

We did the same experiment (Supplementary Figure 11) with an *atg7* mutant (**Supplementary Figure. 12**).

We newly examined the wortmannin effects on *atg2* (**Supplementary Figure 19**) and *atg7* mutants, and showed that wortmannin disturbed ATG18a-GFP targeting on the peroxisome aggregate in both mutants.

We used *atg2* mutants to examine the ATG8 and ATG18a localisation in **Supplementary Figures 13, 15, 16, 17, and 23 (old Supplementary Figures 11, 13, and 19)**, because ATG18a-GFP tightly binds to pexophagy structures in the *atg2* mutant and is easy to track compared to that in the *atg7* mutant.

Performing proteomics analyses on the *atg7* mutant would be another option to address these questions. However, we believe the data from *atg2* should suffice as both *atg2* and *atg7* mutants show a similar effect on peroxisome aggregation in the high-intensity light condition (**Supplementary Figures 20-22; old Supplementary Figure 19**), suggesting that the result for *atg7* might be trivial.

Comment 3. (1) Since most of the conclusions are obtained from confocal imaging studies, the quality of confocal data is of great importance. The authors should distinguish carefully the real punctate associations of (1) ATG18a-GFP or GFP-2×FYVE with organelles from just artifacts caused by occasional space-overlap, which is especially true for the co-localization analysis of those two proteins with chloroplasts shown in Supplementary Figure 23(moved to 34). All of the images to show the co-localizations of RFP-PTS1 labeled peroxisomes with ATG18a-GFP or GFP-2×FYVE or tri-labeling should also include the splitted channels for readers to observe the precise morphology of the structures. Structures in some of the images cannot be clearly visualized in the less-magnified view, e.g. Supplementary

Figure 8b and 12b. Some images would be better presented by showing a large field of view along with the magnified view of some specified structures for readers to understand the frequency, subcellular locations and detailed morphologies, such as supplementary Figure 6 and especially observations of peroxisomes surrounded by vacuolar membranes in Figure 5 and Supplementary Figure 21a.

(2) In addition, the current descriptions about the relative positions of peroxisome aggregates around vacuolar membranes are confusing. Tonoplast and vacuole bubbles were not strictly distinguished and collectively referred to as vacuolar membranes. In addition, whether some peroxisome aggregates appeared within vacuoles, whether those vacuolar-incorporated peroxisome aggregates are enveloped with tonoplast-derived membranes, and the quantification of the numbers of corresponding types of peroxisome aggregates defined by the above features should be clarified, which will help the readers to better understand the occurrence of pexophagy via macroautophagy or microautophagy.

Response 3(1). According to the reviewer's comments, we included the split channel images in the **Source data file**. We added both the magnified and large field images for Supplementary Figures 8b and 12b (moved to 9b and 14, respectively). We also attached a large field of view for Supplementary Figure 6 (protoplast), the detailed morphology of the peroxisomes surrounded by vacuolar membranes in Figure 5, and Supplementary Figure 21a (moved to 26). We also added the split channel images of Supplementary Figure 23 (moved to 31). The list of the additional images is as follows:

[Figures 1a, 2a and e, 3a and e, 5b, e, and g, 6e and f and Supplementary Figures 6b, 7b, 9a and b, 10a-c, 14a-c, 18a and b, 26-29, 30(upper panel), 31]

(2) We changed and unified the phrase to distinguish vacuolar membrane structures enveloping peroxisomes, as “vacuolar cavities”. Besides, we isolated vacuoles to examine whether peroxisome aggregations were inside or outside of vacuole of both the transgenic plant expressing venus-Vam3 or ATG18-GFP in *atg7(p4)* under high-intensity light conditions. We showed that the vacuolar cavities enveloping peroxisomes were generated on the surface of the isolated vacuoles. We quantified the number of peroxisome aggregates enveloped with vacuolar cavities in the *atg7(p4)* mutant and discovered that peroxisome aggregates accumulated in vacuolar cavities in the *atg7(p4)* mutant. Furthermore, we discovered assimilation of peroxisomes into vacuoles in the wild-type cells, but not in the *atg7(p4)* mutant, by the time-lapse imaging analysis of isolated vacuoles. These peroxisome aggregates on the isolated vacuoles accumulated ATG18, indicating that they were in the intermediate phase of the autophagic pathway. These results suggest that peroxisomes were also degraded with micropexophagy. We included additional Figures, **Figure 6 and Supplementary Figs 27-29**, and explanations on **Pages 11–12, lines 236–257** as follows:

[To understand the peroxisome aggregate and vacuolar membrane association, we isolated vacuoles from transgenic plants expressing RFP-PTS1 and venus-VAM3 after exposure to high-intensity light treatment that enhances pexophagy (Fig. 6 and Supplementary Figs. 27-29). RFP fluorescence was observed on the vacuolar surface in the *atg7(p4)*, whereas the fluorescence was observed in the vacuolar lumen of wild-type cells (Fig. 6a–h, Supplementary Figs. 27–29 and Supplementary Videos 18–20). These findings suggest that peroxisome aggregates are attached to the surface of isolated vacuoles in *atg7(p4)* while being further assimilated into vacuoles in the wild type (Fig. 6a-h). The enlarged and time-lapse images revealed that peroxisomes were surrounded by the vacuolar membrane

and assimilated into the vacuole in the wild type (Fig. 6e, f, Supplementary Fig. 27a-d, and Supplementary Videos 18-20). We analysed the frequency of RFP fluorescence surrounded with venus-VAM3 and discovered that approximately 25% of the peroxisome aggregates were surrounded by the vacuolar membrane in the vacuoles isolated from the *atg7(p4)* mutant (Fig 6i). This suggests that although the vacuolar membrane surrounded the peroxisome aggregate, assimilation did not occur in *atg7(p4)*, which is consistent with reduced RFP-fluorescence observed in the vacuolar lumen of the mutant (Supplementary Fig. 27e). Technically, autophagy in which a target for degradation is directly enclosed by the vacuolar membrane and taken up into the vacuole is referred to as microautophagy. Therefore, peroxisome aggregates seem to be degraded via microautophagy, i.e., micropexophagy, under high-light irradiation conditions. We also examined ATG18a localisation to the isolated vacuoles in plants expressing ATG18a-GFP and RFP-PTS1. We observed co-localisation of ATG18a-GFP and RFP-PTS1 on the vacuoles isolated from *atg7(p4)* (Supplementary Figs. 28, 29 and Supplementary Videos 21,22), indicating that the peroxisomes on the vacuolar membranes are in the intermediate part of the autophagic pathway. These results suggested that ATG18 may accumulate during the formation of vacuolar membrane cavities surrounding peroxisome aggregates and that ATG7 is involved in the assimilation of these structures into the vacuoles through the microautophagy process under the high-intensity light condition.]

, and on Page 17, lines 379–380:

[This was also seen in isolated vacuoles (Fig. 6 and Supplementary Figs.27).]

, and on Pages 17–18, lines 389–400:

[We discovered that the vacuolar membrane forms large cavities to surround peroxisome aggregates in *atg7(p4)* under high-intensity light conditions (Fig. 5f–j and Supplementary Fig. 26). Furthermore, some of the peroxisomes and peroxisome aggregates on the surface of the isolated vacuole in wild-type and *atg7(p4)* cells were also surrounded by the vacuolar membrane (Fig. 6 and Supplementary Figs. 27 and Supplementary Videos. 18-21). These direct actions of the vacuolar membrane indicate the involvement of the microautophagy process during the incorporation of degraded peroxisomes into the vacuole. In plants, microautophagy is induced in sucrose-starved root cells ⁵⁸. Microautophagy contributes to the accumulation of anthocyanin aggregate in vacuoles ⁵⁹ and damaged chloroplast degradation under high-intensity light irradiation ^{60,61}. Microautophagic degradation of peroxisomes (micropexophagy) was reported in yeast, where it is accompanied by a micropexophagy-specific apparatus (MIPA) ²², but not yet in plants. Taken together, these findings suggest that micropexophagy is induced following high-intensity light exposure resulting in the degradation of oxidized-peroxisomes and their aggregates.]

Comment 4. The authors performed a lot of quantifications of intensities related to WB and H2-DCF staining. For quantification of WB images in Supplementary Figure 18C and 20f, are the two films exposed at the same time? If not, how can the authors convince readers to believe that the intensity differences were results of signals or just exposure time? This reviewer is also curious that how the H2-DCF staining signals from different leaves of different plants can be scientifically quantified without an internal control to avoid interference of leaf thickness. In addition, the intensities

quantified in Figure 6a indicate some signals were possibly out of the linear dynamic range of the detector and overexposed

Response 4. We captured immunoblot images with a CCD camera using the same parameters for all conditions: exposure time, concentration, and background intensity. We inserted the sentence on **Pages 24, lines 542–543** as follows:

[We captured immunoblot images with a CCD camera by using precisely the same parameters for all conditions (exposure time, contrast, and background intensity).]

In the H₂-DCF-staining, we used the same confocal microscope setting of exposure time and dynamic range across images. Furthermore, we carefully selected samples between plants as the leaf mesophyll cells of similar regions and depths in the leaves. The intensity of fluorescence was 255 in the Image J, therefore we applied this intensity as the maximal value. We inserted an explanatory sentence on **Pages 23–24, lines 527–528** as follows:

[We carefully selected leaf mesophyll cells from similar regions and depths within the leaves and used the same confocal microscope setting of exposure time and dynamic range across images.]

Comment 5. In addition to the localizations of ATG18a-GFP or GFP-2×FYVE on some peroxisomes, where do their other signals mainly appear at the subcellular level. Please add images to show the localization patterns of ATG18a-GFP and GFP-2×FYVE in wild-type cells under normal conditions.

In supplementary figure 15d, some GFP signals of ATG18a-GFP seem to appear on chloroplasts. Is this case normal? Also, the last three panels of the supplementary figure 15d were shown in the bright field mode, while the first one is not. Those data need to be displayed in a consistent manner.

Response 5. We recognized some images have extra fluorescence on chloroplast and the cytosol. We examined the wild-type cells under normal conditions in Figs 1-3, but signals are still there; thus, indicating that these is noise, possibly due to chlorophyll autofluorescence.

As suggested, we changed the image to ensure consistency.

Comment 6. The authors demonstrated that ectopic expression of catalase in *atg2* mutant decreased not only the number of peroxisome aggregates but also the total number of peroxisomes. This result is quite surprising. How can the aggregated peroxisomes can be further degraded in a autophagy-defective mutant?

Response 6. We would like to point out that this is not a degradation issue but rather a matter of peroxisome aggregate progression. We started to expose the *atg2* mutants to high-intensity light after the over-expression of GFP-CAT2 or RFP-CAT2. Therefore, the non-transformant *atg2* plants experienced increased peroxisome numbers and aggregates during the high-intensity light exposure. However, over-expression of GFP-CAT2 or RFP-CAT2 suppresses the progress of peroxisome numbers and aggregates, resulting in the difference between non- and over-expressing plants. We changed the sentence on Page 9, lines 197–199 to make sure is not misleading for the readers as follows:

[We overexpressed GFP-CAT2 or RFP-CAT2 to recover catalase activity and discovered that CAT2 fusion overexpression suppressed the progress of peroxisome numbers and their aggregation in the *atg2* mutant (Supplementary Fig. 23).]

We also modified the sentences on **Page 13, line 284–285,**

[The overexpression of catalase suppresses the increase in peroxisome aggregate number,]

and on **Page 14, line 310–312.**

[The overexpression of catalase suppressed the increase in peroxisome numbers and aggregation in *atg2* leaf cells (Supplementary Fig. 23); thus, suggesting that active catalase would reduce peroxisomal ROS and subsequent peroxisome aggregation in *atg2*.]

Comment 7. As illustrated by the authors, leaf damage under high-intensity light is closely related to catalase inactivation, peroxisome aggregation and excess ROS accumulation. However, although the authors claimed that overexpression of ATG18 significantly suppressing accumulation of ROS within mesophyll cells, the ATG18 OE lines did not show visible alleviated leaf damage under high-intensity light. Besides, the leaf damage in wild-type plants subjected to high light treatment, as shown in Supplementary Figure 17a, seems to be second only to *atg7* mutant, because bleach phenotypes can be both observed in those two plants, but not in other *atg* mutants.

Response 7. We calculated the chlorophyll a + b content in the leaves to assess leaf damage in multiple biological replications (the image shows one example). The result showed less chlorophyll content in *atg2*, *atg5*, and *atg7* than that observed in wild type. We have inserted the results in **Supplementary Figures 20b and 32b** with the additional sentence as follows:

[Leaf damage and chlorophyll degradation were observed in *atg2*, *atg5* and especially *atg7* (Fig. 4a and Supplementary Fig. 20).] (Page 9, lines 189–190)

, and

[The ATG18a-GFP overexpressing plants showed a reduction in ROS accumulation and an increase in chlorophyll content in the leaf cells (Supplementary Fig. 32); thus, suggesting that the ATG18a overexpression reduces ROS accumulation and chloroplast damage.] (Page 12, lines 275–277)

Comment 8. Data involving mitochondrial degradation and chloroplast damage are very preliminary and has little to do with the main scientific question need to be solved in this manuscript, which can be removed from the maintext.

Response 8. We have moved these data to the supplemental figures. We would like to keep the last explanation to compare these with peroxisome degradation.

Comment 9. Supplementary figure 15c and d were not referenced in the main text.

Response 9. We have inserted an additional sentence on Page 14, lines 317–318 as follows,

[We used the ATG18a-GFP form because it complements the mutant phenotype of *atg18a(p2)* (Supplementary Figs, 18c,d), indicating that the fusion protein is functional.]

Comment 10. Internal control for supplementary figure 4a deposited in source data should be assembled at the bottom of the figure. Similar analysis in supplementary figure 20d lacks internal control.

Response 10. Following to the reviewer's comment, we attached the internal control at the bottom of Supplementary Fig 4a. We used the same extraction samples in Supplementary Fig 20d, so we added sentences in the Figure legend.

Comment 11. Figure legend of supplementary figure 4b is not correct.

Response 11. Thank you for pointing out our mistakes. We corrected the Figure legend of Supplementary Figure 4b as follows:

[b, Signal intensity of immunoblotting analysis in (a).]

Comment 12. The TEM image in Supplementary figure 20b is meaningless without being accompanied with the proper control.

Response 12. We deleted the Supplementary Figure 20b.

Comment 13. The pseudocolors for peroxisomes and chloroplast should be different.

Response 13. We added the annotation on the left side of Supplementary Figure 25a.

Comment 14. Splitted images from individual channels for the up panel of Supplementary Figure 22c are missing in the source data.

Response 14. We added the individual channels of Supplementary Figure 22C (moved to 30C) in the source data.

Comment 15. Typo errors in source data:

Supplementary Figure 20f, the labels for antibody of the bottom panel should be anti-SHMT

Supplementary Figure 21b, High light ($100 \mu\text{molm}^{-2}\text{s}^{-1}$) should be High light ($1000 \mu\text{molm}^{-2}\text{s}^{-1}$)

Response 15. We corrected these typos.

Comment 16. The normalization methods for intensity analysis in source data were not consistent:

Raw data for Fig. 4f, Sfig.22b and Sfig.24d were normalized to the wild-type plants, whose mean was set to 1. Raw data for SFig.2e and Sfig.24c were normalized to the wild-type plants, whose mean was not set to 1. Raw data for SFig.4b and SFig.20e were normalized to the wild-type plants, but the value of individuals was set to 1.

Response 16. We modified the normalization method for consistency. The raw data were normalized to the wild-type plants, and the value of individuals was set to 1 in Fig. 4f, SFig. 4b and SFig. 25e (old SFig. 20e) as it was challenging to control the immunoblot setting throughout the experiment, we used wild type cells as a normalizer in each experiment. For the calculation of cell and leaf images, the raw data were normalized to the wild-type plants (or wild type in low light), whose mean was set to 1 in SFig. 2e, SFig. 30b (old SFig. 22b), SFig. 32d (old SFig. 24c) and SFig. 32e (old SFig. 24d) because the images were captured using the same parameters.

Reviewer #3 (Remarks to the Author):

This revised manuscript has addressed most of my concerns and can be considered for the acceptance.